# DYNAMIC NEURAL GRAPH: FACILITATING TEMPORAL DYNAMICS LEARNING IN DEEP WEIGHT SPACE

## ABSTRACT

The rapid advancements in using neural networks as implicit data representations have attracted significant interest in developing machine learning methods that analyze and process the weight spaces of other neural networks. However, efficiently handling these high-dimensional weight spaces remains challenging. Existing methods often overlook the sequential nature of layer-by-layer processing in neural network inference. In this work, we propose a novel approach using dynamic graphs to represent neural network parameters, capturing the temporal dynamics of inference. Our Dynamic Neural Graph Encoder (DNG-Encoder) processes these graphs, preserving the sequential nature of neural processing. Additionally, we also leverage DNG-Encoder to develop INR2JLS for facilitate downstream applications, such as classifying INRs. Our approach demonstrates significant improvements across multiple tasks, surpassing the state-of-the-art INR classification accuracy by approximately 10% on the CIFAR-100-INR. The source code has been made available in the supplementary materials.

## 1 INTRODUCTION

Deep neural networks have demonstrated superb capability in addressing real-world problems in fields such as computer vision, natural language processing, and the natural sciences. While it is generally used for learning patterns from data, recent studies have expanded its scope by treating neural networks themselves as inputs, enabling tasks such as opt imizing networks Metz et al. (2022), predicting the labels of the data encoded in implicit neural representations Dupont et al. (2022), and generating or modifying their weights to alter functionality Schürholt et al. (2022). However, processing these weight spaces presents considerable challenges due to their complex, high-dimensional nature.

To address the difficulty, some existing methods propose to narrow the effective weight space using a restricted training process (Bauer et al. (2023); Dupont et al. (2021); De Luigi et al. (2023)). However, this neglects the crucial permutation symmetry property of the neural network weights , *i.e.,* neurons within a layer can be rearranged without altering the network's function (Hecht-Nielsen (1989)). Overlooking the permutation symmetry can significantly increase the search space for optimal parameters of processing network, resulting in reduced generalization and unsatisfied performance. By observing this, recent works (Navon et al. (2023); Zhou et al. (2024b;a)) build permutation equivariant weight-space models named *neural functionals*. Unfortunately, these methods need manual adaptation for each new architecture, and a single model can only handle one fixed architecture. To encourage process heterogeneous architectures, Kofinas et al. (2024); Lim et al. (2024) introduce to model neural network weights as graph, which links neural network parameters similarly to a computation graph. This methods, while innovative, predominantly employ static graphs. This static representation allows GNNs to process the entire graph in a single pass. However, such an approach overlooks a critical aspect of neural network behavior during inference: the sequential nature of layer-by-layer processing. Neural networks, by design, perform inferences in a temporally ordered manner, where each layer's output serves as the input for the subsequent layer. This sequential dependency suggests that a more natural and effective modeling of neural network parameters could be achieved through dynamic graphs. Unlike static graphs, dynamic graphs evolve over time, capturing the temporal dynamics inherent in the forward pass process.

Motivated by these observations, we propose a novel method that represents neural network parameters as dynamic graphs, namely dynamic neural graph. Leveraging this dynamic graph, we introduce the Dynamic Neural Graph Encoder (DNG-Encoder), a recurrent-like graph neural network designed to process dynamic neural graphs. This approach mirrors the forward propagation mechanism of neural networks, preserving the temporal characteristics of the data flow through the layers. To facilitate downstream applications, we use the DNG-Encoder to develop INR2JLS, a method that learns a joint latent space between deep weights and the original data. This approach provides a more informative latent space compared to previous methods that focused solely on INR weights.

Our contribution can be summarized as follows. First, we introduce the concept of dynamic neural graphs for modeling neural network parameters, capturing the temporal dynamics of the forward pass. Second, we develop a novel RNN-based graph neural network to process these dynamic graphs, effectively imitating the sequential nature of neural network inference. Third, we propose INR2JLS, a technique that maps INR weights into a joint latent space that can benefit developing downstream application. Finally, we show through extensive experiments to validate the effectiveness of our method across three tasks. Notably, the performance of our method improves over the state-of-the-art by 9% and 10% on CIFAR-10 and CIFAR-100 for classifying INR weights.

## 2 PRELIMINARIES ON NEURAL GRAPH

### 2.1 NEURAL NETWORKS AS STATIC NEURAL GRAPH

A recent study (Kofinas et al. (2024)) introduce a novel representation of neural networks, named *neural graphs*, which ensures invariance to neuron symmetries. For example, in an L-layer multilayer perceptron (MLP) $\mathbf{M}$, the weight matrices are denoted as $\{\mathbf{W}^1, \mathbf{W}^2, ..., \mathbf{W}^L\}$, and the biases are denoted as $\{\mathbf{b}^1, \mathbf{b}^2, ..., \mathbf{b}^L\}$. Each weight matrix $\mathbf{W}^l$ and bias $\mathbf{b}^l$ respectively have dimensions $d^l \times d^{l-1}$ and $d^l$. $\mathbf{M}$ can be converted to a neural graph $\mathcal{G} = (\mathbf{V}, \mathbf{E})$, where $\mathbf{V} = \{\mathbf{v}^0, \mathbf{v}^1, ..., \mathbf{v}^L\}$ denotes the set of nodes features, and $\mathbf{E} = \{\mathbf{e}^1, \mathbf{e}^2, ..., \mathbf{e}^L\}$ represents the set of edges features. $\mathbf{v}^l \in \mathbb{R}^{d^l \times d_v}$ and $\mathbf{e}^l \in \mathbb{R}^{d^l \times d^{l-1} \times d_e}$ are the nodes and edges of the $l$-th layer in $\mathcal{G}$, respectively, where $d_v$ and $d_e$ represent the dimensions of the node feature and the edge feature, respectively. In addition, $\mathbf{v}^l$ and $\mathbf{e}^l$ corresponds to neurons at the $l$-th layer of $\mathbf{M}$ and connections between neurons at the $l$-th layer and the $(l-1)$-th layer of $\mathbf{M}$, respectively. Typically, edge feature matrices contain the weights of $\mathbf{M}$, while nodes are constructed using biases. Additionally, the feature matrices may not necessarily be the original weights and biases. Some studies (Kofinas et al. (2024) Zhou et al. (2024b) Zhou et al. (2024a)) have demonstrated improved performance by using frequency representations of $\mathbf{b}^l$ and $\mathbf{W}^l$, such as Random Fourier Features (RFFs) (Rahimi & Recht (2007)). It is worth mentioning that once the neural graph is defined, its structure remains unchanged throughout the analysis or application. According to the definitions of various graphs (West et al. (2001)), this type of neural graph can also be referred to as a static neural graph.

### 2.2 EXPRESSIVITY OF PROCESSING NEURAL GRAPHS WITH GRAPH NEURAL NETWORKS

A pioneer (Navon et al. (2023)) in deep weight processing suggest that if a model can simulate the forward pass process of its input neural network, then it has the ability to exhibit the expressiveness of the input neural network. Following the suggestion, Kofinas et al. (2024) use graph neural networks (GNNs) in the form of message passing neural network (MPNN)[1] (Gilmer et al. (2017)).

Before we start discuss the expressively of GNNs for neural graphs, first, we recall the forward pass of neural network. Given an input $\mathbf{x} \in \mathbb{R}^{d^0}$ for $\mathbf{M}$, an $i$-th activation $a_i^1$ of the first layer in $\mathbf{M}$ can be obtained as follows:

$$a_i^1 = \sigma(\mathbf{b}_i^1 + \sum_j \mathbf{W}_{ij}^1 \mathbf{x}_j), \tag{1}$$

where $\sigma$ is an activation function.

Given a MPNN with $K \in \mathbb{R}^1$ layers in total, we have $\mathbf{v}^l(k)$ and $\mathbf{e}^l(k)$ to represent the nodes and edges at the $l$-th layer of the neural graph under the processing of $k$-th layer of MPNN . Usually, we

---

[1]Please refer to Appendix B in Kofinas et al. (2024) for detailed discussions.

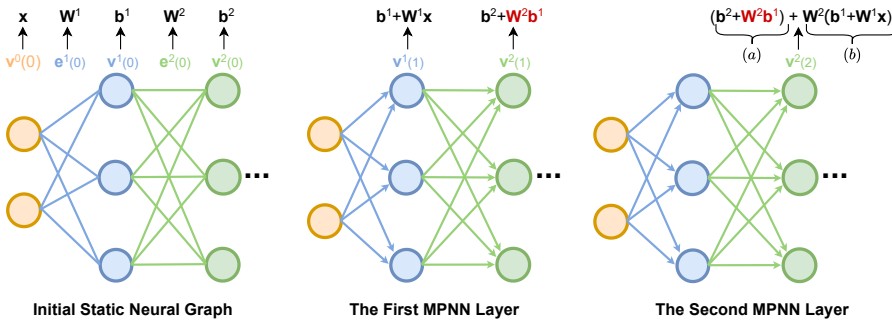

Figure 1: An illustration of the limitations in processing static neural graphs. As the processing of the static neural graph goes into deep layer, the updated nodes may contain additional information that is not desired, such as the $\mathbf{W}^2\mathbf{b}^1$ in (a).

have $K = L$. The message-passing process of the first MPNN layer can be written as:

$$\mathbf{v}_i^1(1) = \phi_u^1(\mathbf{v}_i^1(0), \sum_{j \in N_i} \phi_m^1\left(\mathbf{v}_i^1(0), \mathbf{e}_{ij}^1(0), \mathbf{v}_j^0(0)\right), \qquad (2)$$

where $i$ and $j$ represent the index of the target node and the source node. $N_i$ represents the neighbors of node $\mathbf{v}_i$. For simplicity, here we assume these indexes match that of network parameter.

By comparing the Equation 1 and 2, the authors in Kofinas et al. (2024) emphasize that the MPNN can approximate the feed-forward procedures on input networks of the first layer in MPNN as follows. Please be noted that $\mathbf{v}_i^1(0)$, $\mathbf{e}_{ij}^1(0)$, and $\mathbf{v}_j^0(0)$ are the initial representations directly derived from $\mathbf{b}_i^1$, $\mathbf{W}_{ij}^1$, and $\mathbf{x}_j$, respectively. The function $\phi_m^1$ is the message function of the first MPNN layer, capable of approximating the scalar product $\mathbf{W}_{ij}^1\mathbf{x}_j$. The function $\phi_u^1$ represents the node update function of the first MPNN layer. It can easily approximate the operation of adding $\mathbf{b}_i^1$ to $\mathbf{W}_{ij}^1\mathbf{x}_j$ and applying the activation function $\sigma$. In this way, the MPNN is capable of approximating feed-forward procedures of input networks.

## 2.3 LIMITATIONS OF STATIC NEURAL GRAPHS

As is well known, neural networks perform inference sequentially, where each subsequent layer requires the activation produced by all preceding layers to function correctly. In the static neural graph (Kofinas et al. (2024)), a node only connects to its adjacent nodes, either in the preceding or succeeding layers. Therefore, when updating a node, it can only reference information from its neighboring nodes. This updating rule contradicts the sequential updating pattern of neural networks. Moreover, we also identify a significant issue when employing multi-layer graph neural networks discussed below.

Following the message-passing process of the first MPNN layer on the first set of MLP nodes in Equation 2, the message-passing process of the first MPNN layer on the second set of MLP nodes is as $\mathbf{v}_i^2(1) = \phi_u^1(\mathbf{v}_i^2(0), \sum_{j \in N_i} \phi_m^1\left(\mathbf{v}_i^2(0), \mathbf{e}_{ij}^2(0), \mathbf{v}_j^1(0)\right))$. Since this graph update uses the same $\phi_u^1$ and $\phi_m^1$ as those in Equation 2, according to the expressivity of GNN, each node $\mathbf{v}_i^2(1)$ in the second layer of the MLP should be updated to $\mathbf{b}_i^2 + \mathbf{W}_i^2\mathbf{b}^1$. For clarity, we omit including $\sigma$.

Now, we move to the second layer of MPNN. When we examine the update of node $\mathbf{v}_i^2(2)$ in this layer, we have: $\mathbf{v}_i^2(2) = \phi_u^2(\mathbf{v}_i^2(1), \sum_{j \in N_i} \phi_m^2(\mathbf{v}_i^2(1), \mathbf{e}_{ij}^2(1), \mathbf{v}_j^1(1)))$. Again, following the expressivity of GNN, we can assume this calculation completes: $\underbrace{\mathbf{b}_i^2 + \mathbf{W}_i^2\mathbf{b}^1}_{(a)} + \underbrace{\mathbf{W}_i^2(\mathbf{b}_i^1 + \mathbf{W}_i^1\mathbf{x})}_{(b)}$, where part (b) is approximated by $\phi_m^2$, and adding part (a) to (b) is done by $\phi_u^2$. However, if we strictly follows the forward pass of neural network, the desired computation should be: $\underbrace{\mathbf{b}_i^2}_{(c)} + \underbrace{\mathbf{W}_i^2(\mathbf{b}_i^1 + \mathbf{W}_i^1\mathbf{x})}_{(d)}$. In this case, besides the approximation of addition operation, $\phi_u^2$ needs to do additional work to extract $\mathbf{b}_i^2$ from the summarized result $\mathbf{b}_i^2 + \mathbf{W}_i^2\mathbf{b}^1$. This may seem like an easy task, but technically it is not.

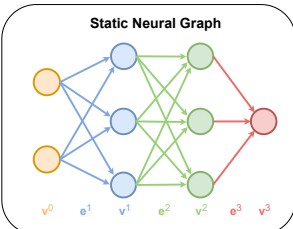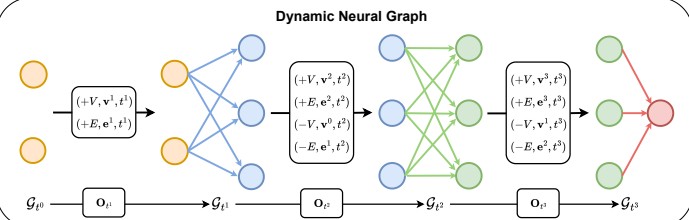

Figure 2: Left: static neural graph. Right: dynamic neural graph. A static neural graph has a fixed structure and set of connections, while dynamic neural graph evolves over time, with changes in its structure and node connections.

Extracting $\mathbf{b}_i^2$ from $\mathbf{b}_i^2 + \mathbf{W}_i^2 \mathbf{b}^1$ constitutes a typical inverse problem, which is inherently ill-posed and challenging to solve. Consequently, training a network to perform this task can lead to difficulties in convergence and may result in suboptimal solutions. To facilitate understanding, we show the problem in Figure 1.

## 3 NEURAL NETWORKS AS DYNAMIC GRAPH

To address the above limitations of static neural graphs, we propose converting the input neural network into dynamic graphs. This conversion incorporates the inherent temporal processing characteristics of neural networks directly into the graph structure, enabling subsequent models to effectively capture and utilize these temporal relations during graph processing. In the following, we discuss the conversion of both MLPs and CNNs into dynamic neural graphs.

### 3.1 MLPS AS DYNAMIC NEURAL GRAPHS

We define a dynamic graph converted from an $L$-layer MLP $\mathbf{M}$ as a *dynamic neural graph* $\mathcal{G}_T = (\mathcal{G}_{t^0}, \mathbf{O}_{[t^1:t^L]})$, where $\mathcal{G}_{t^0}$ only contains $\mathbf{v}^0$ that corresponds to inputs of $\mathbf{M}$. The definitions of $\mathbf{v}^l$ and $\mathbf{e}^l$ in the dynamic neural graph are the same as those in the static neural graph from Section 2.1. Since the inputs to neural networks are not fixed, we treat them as learnable vectors. To keep the dimensions of all node embeddings consistent, we set their dimensions to be the same as the dimensions of the embeddings of other nodes. We simulate the forward pass process of $\mathbf{M}$ by defining the graph update event $\mathbf{O}$. We define four graph operations, *i.e.,* edge addition $(+E)$, edge deletion $(-E)$, node addition $(+V)$ and node deletion $(-V)$. Specifically, a graph update event $\mathbf{O}_{t^l}$ at time $t^l$ is:

$$\mathbf{O}_{t^l} = \begin{cases} \{(+V, \mathbf{v}^l, t^l), (+E, \mathbf{e}^l, t^l)\} & \text{if } l = 1, \\ \{(+V, \mathbf{v}^l, t^l), (+E, \mathbf{e}^l, t^l), (-V, \mathbf{v}^{l-2}, t^l), (-E, \mathbf{e}^{l-1}, t^l)\} & \text{if } 1 < l \leq L, \end{cases} \quad (3)$$

where $(+V, \mathbf{v}^l, t^l)$ denotes adding the nodes $\mathbf{v}^l$ to $\mathcal{G}_{t^l}$ at timestamp $t^l$. $(+E, \mathbf{e}^l, t^l)$ represents adding edges $\mathbf{e}^l$ to to $\mathcal{G}_{t^l}$ at timestamp $t^l$. The edges $\mathbf{e}^l$ connect nodes $\mathbf{v}^{l-1}$ to the newly added nodes $\mathbf{v}^l$. When $t^1 < t^l \leq t^L$, we delete nodes $\mathbf{v}^{l-2}$ and edges $\mathbf{e}^{l-2}$, and adding incoming nodes and edges.

Similar to many previous approaches to process weight space parameters (Zhou et al. (2024b)Zhou et al. (2024a)Kofinas et al. (2024)), we use the Random Fourier Features (RFFs) of weights $\mathbf{W}^l$ and biases $\mathbf{b}^l$ in $\mathbf{M}$ to initialize $\mathbf{v}^l$ and $\mathbf{e}^l$. By the above definition of $\mathbf{O}$, the snapshot $\mathcal{G}_{t^l} = (\{\mathbf{v}^{l-1}, \mathbf{v}^l\}, \mathbf{e}^l)$ at timestamp $t^l$ can be considered a fully connected static bipartite directed graph (Bang-Jensen & Gutin (2008)). This graph consists of two sets of nodes, $\mathbf{v}^{l-1}$ and $\mathbf{v}^l$, with the direction of edges $\mathbf{e}^l$ from nodes $\mathbf{v}^{l-1}$ to nodes $\mathbf{v}^l$. In this way, the structure of $\mathcal{G}_{t^l}$ complies with the topology of the $l$-th forward pass step of $\mathbf{M}$. To better illustrate the procedure of converting an MLP to a dynamic neural graph, we show an example in Figure 2.

Besides, Kofinas et al. (2024) prove that the natural symmetries in the graphs align with neuron permutation symmetries in neural networks. For example, permuting the nodes of the neural graph adjusts the adjacency matrix in a way that connections between same neurons remain the same. In our

dynamic neural graph, this still holds as our graph operations does not change the original connection between neurons.

## 3.2 CNNs as Dynamic Neural Graphs

A convolutional neural network (CNN) typically consists of convolutional layers and linear layers [2]. We propose a method to convert convolutional layers and linear layers in CNNs to modules in dynamic neural graphs.

A 2D convolutional layer at the $l$-th level of the CNN includes filter $\mathbf{W}^l \in \mathbb{R}^{c^l \times c^{l-1} \times h^l \times w^l}$ and bias $\mathbf{b}^l \in \mathbb{R}^{c^l}$, where $c^{l-1}$ and $c^l$ respectively denote the depth and the number of filters in the $l$-th convolutional layer. Typically, $c^{l-1}$ strictly matches the number of input channels, and $c^l$ controls the number of output channels. $h^l$ and $w^l$ represent the width and height of the kernel.

Naturally, we can treat the biases as node features similarly to how we handle them in an MLP. However, kernels cannot be treated as edge features in the same manner because their spatial dimensions differ from those of MLP weights. To address this problem, Kofinas et al. (2024) propose to flatten spatial kernels to vectors, which forms $c^{l-1} \times c^l$ edges from the $l$-th convolutional layer with each edge represented by a channel of kernel. Considering the fact that the kernel size might be different across convolutional layers, to ensure the consistency in the dimension of edge features, they further propose to pad the vectors to the maximum of $h^l \times w^l$.

In contrast, we regard each weight scalar as an independent edge. Specifically, we construct edges between nodes of adjacent layers, $\mathbf{v}^{l-1}$ and $\mathbf{v}^l$, by a number of $c^{l-1} \times c^l \times h^l \times w^l$ edges, with each edge corresponding to a scalar in $\mathbf{W}^l$ and each pair of nodes is connected by $h^l \times w^l$ edges. To maintain consistency in the number of edges between each pair of nodes within the dynamic neural graph, we perform padding by adding additional zero edges between a pair of nodes to reach the maximum of $h^l \times w^l$.

## 4 Learning Invariant Latent Space on Dynamic Neural Graph

### 4.1 RNN-based Graph Neural Network

As we discuss above, modeling neural networks as dynamic graphs aligns more closely with the forward-pass nature of neural network inference. Notably, Rossi et al. introduced an encoder framework called Temporal Graph Neural Network (TGN), which has demonstrated considerable potential in enabling dynamic neural graphs to more effectively approximate the forward pass process of neural networks. They define the states of nodes in the graph at different timestamps as their *memories*, which continuously update based on observed events relevant to them. Consequently, the encoder can produce *temporal embeddings* from the node memories at any timestamp. Typically, in TGN framework, Message Function and Message Aggregator modules generate messages received by the node, Memory Updater module updates the node memory based on its received messages, and Embedding Module generate the temporal embedding of the node from its memory. Since the Memory Updater module in TGN employs a RNN to update node memory, it can be categorized as an *RNN-based dynamic graph encoder* (Kazemi et al. (2020)). Following the setting of TGN, we customize an RNN-based dynamic graph encoder for processing dynamic neural graphs. We call it *Dynamic Neural Graph Encoder* (*DNG-Encoder*).

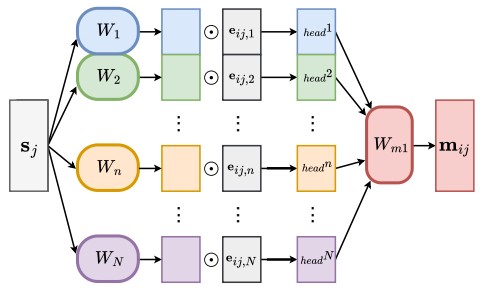

Figure 3: An illustration of multi-head message function, formallized in Equation 5 .

**Message Passing.** Inspired by the process of multiplying activations by weights in a DNN, we use the linear complexity *conditional scaling* mechanism from FiLM (Perez et al. (2018)) to define the Message Function for the DNG-Encoder. Notably, unlike the original FiLM and the method

---

[2]Since the CNNs utilized in our experiments do not incorporate the flattening layer and residual connections, we discuss them in Appendix G.1.

proposed by Kofinas et al. (2024), we do not include information about the target nodes and the shift module in our operation.

Below we present the Message Function for the case where there is only one edge between a pair of nodes (*i.e.,* for the dynamic neural graph converted by an MLP):

$$\mathbf{m}_i(t^l) = \phi_m^{t^l}(\mathbf{s}_j(t^l-), \mathbf{e}_{ij}(t^l)) = \sum_{j \in \mathbf{N}_i} W_{m1}^{t^l} \mathbf{e}_{ij}(t^l) \odot W_{m2}^{t^l} \mathbf{s}_j(t^l-) \tag{4}$$

where $\mathbf{e}_{ij}(t^l)$ denotes the edge between the target node $\mathbf{v}_i$ and the source node $\mathbf{v}_j$ at time $t^l$. $\mathbf{s}_j(t^l-)$ represent the memories of $\mathbf{v}_j$ just before time $t^l$. $W_{m1}^{t^l}$ and $W_{m2}^{t^l}$ are two linear layers to perform linear tranformation.

For the case of a pair of nodes connected by multiple edges, saying $N$ edges (i.e. for the dynamic neural graph converted by a CNN), we map the source node memory to $N$ heads. Each head interacts with one edge to generate multiple messages. Finally, we merge these messages through an MLP $\phi_h$:

$$\mathbf{m}_i(t^l) = \sum_{j \in \mathbf{N}_i} \phi_h^{t^l} \left( \text{Concat} \left( head_{ij}^1(t^l), \ldots, head_{ij}^N(t^l) \right) \right),$$
$$\text{where} \quad head_{ij}^n(t^l) = W_{m1}^{t^l} \mathbf{e}_{ij,n}(t^l) \odot W_n^{t^l} \mathbf{s}_j(t^l-). \tag{5}$$

We illustrate the multi-head message passing function in Figure 3. It is worth noting that a similar operation, referred to as "multiple towers" in Gilmer et al. (2017), was proposed to address the computational challenges that arise when the dimensionality of node embeddings becomes excessively large. In contrast, our multi-head message function is primarily designed to ensure that a source node can transmit $N$ distinct messages to a target node through $N$ edges, thereby more effectively simulating the forward propagation process of a convolutional layer.

**Recurrent Memory Updating.** Recent works, such as DAGNN (Thost & Chen (2021)) and GHN (Zhang et al. (2018)), have demonstrated the effectiveness of using Gated Recurrent Units (GRUs) to capture complex dependencies during node representation updates. Inspired by these approaches, we similarly employ GRUs to update the memory of target nodes $\mathbf{v}_i^l$, enabling the capture of sequential dependencies between the layers of neural networks:

$$\mathbf{s}_i(t^l) = \phi_u^{t^l} \left( \mathbf{m}_i(t^l), \mathbf{v}_i(t^l) \right) = \text{GRU} \left( \mathbf{m}_i(t^l), \mathbf{v}_i(t^l) \right), \tag{6}$$

Since the proposed DNG-Encoder processes the dynamic graph in a sequential manner, it differs from the MPNN used for static graphs (Kofinas et al. (2024)). Therefore, we can omit the introduction of the "inverse problem" discussed in Section 2.3. For a comprehensive explanation of how our method addresses these limitations, please refer to Appendix D.

It is worth mentioning that we do not update edge representations using the DNG-Encoder. For our dynamic neural graph framework, each edge is only utilized for message passing under a specific timestamp. Other than this timestamp, the edge is not included in the graph structure, meaning the same edge is not reused for multiple message-passing steps. For example, in Figure 2 (right), the edges in $\mathcal{G}_{t^1}$ are not present in the following processing timestamp. Therefore, updating edge features does not affect message passing process of our model. Additionally, the introduced dynamic neural graph allows us to simplify the temporal graph neural network framework by removing the message aggregator and embedding module, enhancing computational efficiency. The main reason is that each node in our dynamic neural network interacts only with the graph features at the current time, avoiding the memory staleness issue identified in Kazemi et al. (2020), which is typically managed by the embedding module. This property also eliminates the need for a memory aggregator, usually employed to integrate messages received by a node from distinct events at various timestamps.

## 5 INR2JLS: LEARNING A JOINT LATENT SPACE FROM DATA AND WEIGHTS

Recent works such as INR2VEC (De Luigi et al. (2023)) and INR2ARRAY (Zhou et al. (2024b)) have made great progress on the learning of latent representations of implicit neural representations (INRs) for downstream tasks like classification. While INR2VEC requires shared initialization of input INRs, INR2ARRAY make substantial improvements so that it can work with randomly initialized INRs.

Besides, both INR2VEC and INR2ARRAY map INR weights into a latent space using an encoder-decoder setup. Similar to the image reconstruction, they propose to reconstruct the INR weights that can function equivalently to the input INRs. As we previously discussed, deep weights are high-dimensional in nature and hard to handle. Similarly, generating INR weights from a latent space can be extremely difficult, potentially increasing the optimization difficulty.

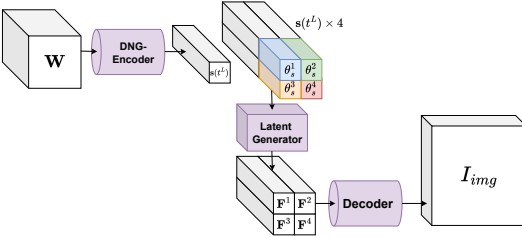

Figure 4: An overview of INR2JLS.

The optimization difficulty can lead to poor network convergence and a suboptimal latent space.

To this end, we introduce INR2JLS, a novel framework that supports randomly initialized INRs and provides a more informative latent space. Compared with the INR2VEC and INR2ARRAY, the main innovation of our INR2JLS is that we introduce a joint latent space between deep weights and the original data. To be specific, INR2JLS utilizes an encoder-decoder architecture to map INR weights to a latent representation capable of capturing both spatial and semantic information inherent in the original image. To achieve this, we do not decode the latent representations back to INR weights. Instead, we decode the latent representation to the original image represented by the input INR. We provide an overview of INR2JLS in Figure 4.

Our encoder, $\text{ENC}_\theta$, is built with the *DNG-Encoder* defined in 4.1 and a *Latent Generator* $\phi_g$. By transforming the input INR weights to dynamic neural graphs, we first employ DNG-Encoder to recurrently process the graph and obtain the last recurrent memory $\mathbf{s}(t^L)$. As discussed earlier, $\mathbf{s}(t^L)$ should inherit all the information contained in a complete forward pass of INR. However, directly decoding an image from $\mathbf{s}(t^L)$ is extremely difficult. One potential reason is that $\mathbf{s}(t^L)$ may store very little spatial information of the original image. Inspired by the positional encoding technique, we introduce a set of learnable spatial vectors $\{\theta_s^1, ..., \theta_s^N\}$, where $N = h_s \times w_s$ denotes the dimensional of latent representation. For generating a single latent vector, we have:

$$\mathbf{F}^n = \phi_g(\text{Concat}(\mathbf{s}(t^L), \theta_s^n)) \tag{7}$$

where $\mathbf{F}^n \in \mathbb{R}^d$ is the feature vector at the index $n$. We conduct Equation 7 over all the set $\{\theta_s^1, ..., \theta_s^N\}$. In this way, we have a set of feature vectors $\{\mathbf{F}^1, ..., \mathbf{F}^N\}$. Then, we reshape the set of feature vector to form a 3D feature map $\mathbf{F} \in \mathbb{R}^{h_s \times w_s \times d}$. Each latent vector in the $\mathbf{F}$ corresponds to a spatial area in the original image.

Finally, we use transposed convolutional layers as a decoder $\text{DEC}_\theta$ to decode $\mathbf{F}$. The objective is to minimize the difference between the decoded outputs and the original images $I_{img}$. We use MSE as the loss function:

$$\mathcal{L}(\theta, \mathbf{W}) = \text{MSE}(\text{DEC}_\theta(\mathbf{F}), I_{img})), \tag{8}$$
$$\text{where} \quad \mathbf{F} = \text{ENC}_\theta(\mathbf{W}) \tag{9}$$

In image processing, implementing data augmentations is a common method to improve the generalization of trained networks. Current data augmentation techniques for deep space processing either encode augmented images into implicit neural representations (INRs) (Kofinas et al. (2024); Zhou et al. (2024a)) or directly modify the INR weights (Shamsian et al. (2024)). With the proposed INR2JLS, we introduce a distinct augmentation method that can be easily implemented in our framework. Specifically, we generate different views of the original images using the decoder. By encouraging the INR2JLS to generate diverse views of the image $I_{img}$, the model can learn representations $\mathbf{F}_{aug}$ that are more robust and invariant to such transformations. A more detailed discussion can be found in Appendix H.4.

## 6 EXPERIMENTS ON DOWNSTREAM APPLICATIONS

To demonstrate the effectiveness of the proposed approach, we conduct a comprehensive evaluation of our method in accordance with Kofinas et al. (2024); Navon et al. (2023); Zhou et al. (2024a). This evaluation involves a series of experiments across multiple tasks, each designed to utilize deep neural network weights as inputs. Specifically, these tasks include: (1) classifying INRs; (2) manipulating

Table 1: Test accuracy (%) for the INR classification task utilizing 10 views of input INRs as data augmentation across various datasets. #Params denotes the number of parameters required in the inference. We adopt 64 probe features for NG-GNN and NG-T, and expanding their size to match a comparable number of inference parameters as our model.

|  | #Params | MNIST | FashionMNIST | CIFAR-10 | CIFAR-100 |
|---|---|---|---|---|---|
| NFN | $\sim$135M | $92.9_{\pm 0.38}$ | $75.6_{\pm 1.07}$ | $46.6_{\pm 0.13}$ | $20.55_{\pm 0.93}$ |
| INR2ARRAY(NFT) | $\sim$59M | $98.5_{\pm 0.00}$ | $79.3_{\pm 0.00}$ | $63.4_{\pm 0.00}$ | $31.30_{\pm 0.04}$ |
| NG-GNN | $\sim$6M | $97.3_{\pm 0.02}$ | $86.53_{\pm 0.58}$ | $55.11_{\pm 1.43}$ | $26.50_{\pm 1.32}$ |
| NG-T | $\sim$6M | $96.83_{\pm 0.06}$ | $85.24_{\pm 0.13}$ | $57.7_{\pm 0.36}$ | $31.65_{\pm 0.28}$ |
| INR2JLS(ours) | $\sim$6M | $\mathbf{98.6}_{\pm 0.01}$ | $\mathbf{90.6}_{\pm 0.07}$ | $\mathbf{73.2}_{\pm 0.28}$ | $\mathbf{42.4}_{\pm 0.32}$ |

INR weights to facilitate image transformations; and (3) assessing the generalization capabilities of CNN classifiers by analyzing their weights. We compare the proposed one with several state-of-the-arts, including NFN (Zhou et al. (2024a)), NFT (Zhou et al. (2024b)) and NG-GNN/NG-T (Kofinas et al. (2024)). More implementation details of all experiments are provided in Appendix H.

## 6.1 CLASSIFYING INRS WITH INR2JLS

**Experiment Setup.** There are mainly two steps in order to use our proposed framework for INR classification. First, we utilize the INR2JLS framework introduced in Section 5, which uses DNG-Encoder proposed in Section 4.1 along with a proposed augmentation strategy to generate a permutation-invariant implicit feature map, $F_{aug}$, that captures diverse semantic information from images via reconstruction. Our augmentation involves five transformations, including clockwise rotations of 90, 180 and 270 degrees, as well as horizontal and vertical flips. This increases the channels of $\mathbf{F}_{aug}$ to six times of a single latent feature $\mathbf{F}$, represented as $\mathbf{F}_{aug} \in \mathbb{R}^{h_s \times w_s \times 6d}$.

Second, we keep the DNG-Encoder and the Latent Generator fixed and add additional classification CNN that takes $\mathbf{F}_{aug}$ as inputs. During training, given a set of INR weights and their corresponding label, the DNG-Encoder processes the INR weights to produce $\mathbf{F}_{aug}$ . Then $\mathbf{F}_{aug}$ is used as input to the classification CNN, and we optimize the CNN using cross-entropy loss to learn a mapping from latent space to label space. It is worth emphasizing that during the training for classifying INRs, we do not directly optimize our encoder to encourage it extracting label-relevant features from input INRs. Instead, we employ a pre-trained encoder, which has learned in a self-supervised manner. In this case, the performance of classification can indicate the quality of the learned latent space.

We conduct the classification task on the public datasets introduced by Zhou et al. (2024a), *i.e.,* MNIST INRs dataset, FashionMNIST INRs dataset, and CIFAR-10 INRs dataset. Besides, to further compare our method with the state-of-the-arts, we perform the INR classification on the more challenging CIFAR-100 INRs dataset. Each image in all datasets contains INRs with 10 views.

**Comparison Results.** Table 1 presents a comparison between our method and several state-of-the-art approaches on the test sets of the four aforementioned datasets. Our method consistently exhibits superior performance compared to all other approaches. Notably, as the difficulty of the dataset increases, our method not only outperforms the best existing method but does so by an increasingly larger margin. Particularly noteworthy is the performance on the challenging CIFAR-10 and CIFAR-100 datasets, where our method surpasses other models by at least 9% and 10%, respectively. This indicates that our method effectively extracts representative features from deep weight spaces.

## 6.2 EDITING INRS

The objective of the INR editing task is to directly manipulate the weight space of the INR. The manipulation can result in a desired transformation of the encoded image, such as image dilation or erosion. Previous approaches utilized the permutation-equivariance property of the model to learn a bias $\Delta(\mathbf{W})$ from the weight space parameters $\mathbf{W}$, which can be added to $\mathbf{W}$ to produce a modified weight space parameters $\mathbf{W}'$. Finally, the transformed image can be generated using the INR $\mathbf{W}'$.

However, as discussed in Section 4, our method does not require updating edge representation for an improved implementation efficiency. Therefore, the typical method of modifying $\mathbf{W}$ by adding a learned offset $\Delta(\mathbf{W})$ is not directly applicable in our framework. To facilitate editing, we employ

Table 2: Test MSE loss (lower is better) for MNIST erosion/dilation/gradient, and FashionMNIST gradient tasks with 10 views of input INRs as data augmentation.

|  | NFN (HNP) | NFN (NP) | NFT | NG-GNN | NG-T | DNG-Encoder (ours) |
|---|---|---|---|---|---|---|
| MNIST (erosion) | $0.0217_{\pm 0.0004}$ | $0.0214_{\pm 0.0007}$ | $0.0194_{\pm 0.0002}$ | $0.0417_{\pm 0.0004}$ | $0.0193_{\pm 0.0007}$ | $\mathbf{0.0071}_{\pm 0.0004}$ |
| MNIST (dilation) | $0.0628_{\pm 0.0009}$ | $0.0628_{\pm 0.0001}$ | $0.0510_{\pm 0.0004}$ | $0.0547_{\pm 0.0003}$ | $0.0486_{\pm 0.0003}$ | $\mathbf{0.0125}_{\pm 0.0005}$ |
| MNIST (gradient) | $0.0541_{\pm 0.0011}$ | $0.0537_{\pm 0.0006}$ | $0.0484_{\pm 0.0007}$ | $0.0907_{\pm 0.0020}$ | $0.0484_{\pm 0.0004}$ | $\mathbf{0.0153}_{\pm 0.0007}$ |
| FashionMNIST (gradient) | $0.0843_{\pm 0.0020}$ | $0.0857_{\pm 0.0001}$ | $0.0800_{\pm 0.0002}$ | $0.1002_{\pm 0.0013}$ | $0.0777_{\pm 0.0006}$ | $\mathbf{0.0434}_{\pm 0.0015}$ |

Table 3: Kendall's rank correlation coefficient $\tau$ for various models in predicting the generalization performance of CNN classifiers on the CIFAR-10-GS and SVHN-GS datasets.

|  | NFN(HNP) | NFN(NP) | NFT | NG-GNN | NG-T | DNG-Encoder(ours) |
|---|---|---|---|---|---|---|
| CIFAR-10-GS | $0.934_{\pm 0.001}$ | $0.922_{\pm 0.001}$ | $0.926_{\pm 0.001}$ | $0.930_{\pm 0.001}$ | $0.935_{\pm 0.000}$ | $\mathbf{0.936}_{\pm 0.000}$ |
| SVHN-GS | $\mathbf{0.931}_{\pm 0.005}$ | $0.856_{\pm 0.001}$ | $0.858_{\pm 0.000}$ | - | - | $0.867_{\pm 0.002}$ |

a more efficient approach by using the INR2JLS framework with DNG-Encoder directly on $\mathbf{W}$ to generate the desired transformed images.

Table 2 presents a comparison of the performance of our model with previous models on tasks of MNIST erosion, MNIST dilation, MNIST gradient, and FashionMNIST gradient. It is evident that our model outperforms other models significantly across all tasks.

### 6.3 PREDICTING CNN CLASSIFIER GENERALIZATION

The above experiments evaluate the capability of our model to handle dynamic neural graphs built from the weight space of INRs. Here, we aim to evaluate the performance of our method on processing dynamic neural graphs built with CNN architecture. It is worth emphasizing that the objective of this experiment is to predict the test accuracy of a trained CNN classifier using its parameters. Following Zhou et al. (2024a), we conduct this experiment on the Small CNN Zoo (Unterthiner et al. (2020)) dataset. This dataset contains thousands of CNNs trained on the public image classification datasets. We follow Zhou et al. (2024a) to evaluate the CNNs trained on CIFAR-10-GS and SVHN-GS datasets.

We employ the DNG-Encoder to process the dynamic neural graph derived from the input CNNs. Subsequently, we add an MLP to map the recurrent memory $\mathbf{s}(t^L)$ of the last graph layer at the final timestamp to the predicted test accuracy of the CNN. We use binary cross-entropy loss in the training.

Table 3 shows the test performance of different models on the CIFAR-10-GS and SVHN-GS datasets using the rank correlation $\tau$ (Kendall (1938)) as the metric. It can be observed that our model outperforms all other methods on the CIFAR-10-GS dataset. However, we underperforms the NFN(HNP) model on the SVHN-GS dataset. As suggested in Zhou et al. (2024b), HNP designs may be naturally better suited to this task.

## 7 FURTHER EMPIRICAL ANALYSIS

To further demonstrate the effectiveness of each module/components in the proposed INR2JLS, we present a comprehensive analysis below. Additional empirical results on positional encoding and non-linearity embedding are provided in Appendix I.

**Analysis of the Data Augmentation.** We conduct an analysis on the effectiveness of the augmentation strategy introduced in Section 5. By comparing the third and fifth row in Table 4, it can be found that the rotation and flip augmentation can help improve the classification accuracy significantly. For example, on complex datasets such as CIFAR-10 and CIFAR-100, using rotation and flip augmentations can improve over baseline by approximately 7% and 9%.

**The Importance of Image Reconstruction in the INR2JLS.** First, we conduct an experiment to compare the performance of two frameworks, our proposed image reconstruction framework (INR2JLS), and the INR-weight reconstruction framework (INR-INR), on the INR classification task. For the INR-INR framework, we employ the DNG-Encoder but use two MLPs to map the node memory, generated by the encoder, to the weights and biases of an INR. Subsequently, we adopt a methodology same as the NFT (Zhou et al. (2024b)) to compute the loss between the image

Table 4: Test accuracy (%) of INR classification using INR2JLS, with/without data augmentation.

|  | MNIST | FashionMNIST | CIFAR-10 | CIFAR-100 |
|---|---|---|---|---|
| No Augmentation | $98.5_{\pm 0.00}$ | $89.5_{\pm 0.07}$ | $66.4_{\pm 0.19}$ | $32.9_{\pm 0.31}$ |
| Adding Noise Augmentation | $98.4_{\pm 0.01}$ | $89.5_{\pm 0.06}$ | $67.3_{\pm 0.38}$ | $33.0_{\pm 0.24}$ |
| Rotation&Flip | $\mathbf{98.6}_{\pm 0.01}$ | $\mathbf{90.6}_{\pm 0.07}$ | $\mathbf{73.2}_{\pm 0.28}$ | $\mathbf{42.4}_{\pm 0.32}$ |

Table 5: Top: Ablation study on the image reconstruction in the INR2JLS. Bottom: Ablation study on the importance of different modules (DNG-Encoder, Latent Generator) in INR2JLS. Results are shown in classification accuracy (%).

|  | Method | MNIST | FashionMNIST | CIFAR-10 | CIFAR-100 |
|---|---|---|---|---|---|
| Recon. Study | INR2JLS (Ours) | $98.6_{\pm 0.01}$ | $\mathbf{90.6}_{\pm 0.07}$ | $\mathbf{73.2}_{\pm 0.28}$ | $\mathbf{42.4}_{\pm 0.32}$ |
| | INR-INR | $98.6_{\pm 0.08}$ | $88.3_{\pm 0.04}$ | $56.3_{\pm 0.25}$ | $30.6_{\pm 0.16}$ |
| Modules' Study | DNG-Encoder | $96.6_{\pm 0.09}$ | $78.4_{\pm 0.61}$ | $54.0_{\pm 0.07}$ | $25.7_{\pm 0.12}$ |
| | INR2JLS w/o Latent Generator | $98.4_{\pm 0.08}$ | $88.9_{\pm 0.28}$ | $54.5_{\pm 0.51}$ | $28.1_{\pm 0.43}$ |
| | INR2JLS (Ours) | $\mathbf{98.6}_{\pm 0.01}$ | $\mathbf{90.6}_{\pm 0.07}$ | $\mathbf{73.2}_{\pm 0.28}$ | $\mathbf{42.4}_{\pm 0.32}$ |

obtained by the reconstructed INR and the original image. Finally, an MLP is employed to classify the INR based on the node memory output from the pretrained encoder. Table 5 (Top) shows the classification performance of the two frameworks on INR datasets. Our INR2JLS outperforms INR-INR, highlighting the significance of learning a joint space between INR and the original images.

**Ablation Study of Key Components in INR2JLS.** We here conduct ablation experiments to assess the individual contributions of the two modules (Decoder and Latent Generator) within the INR2JLS framework towards enhancing performance in the INR classification task. In the first ablation experiment, we remove the decoder from INR2JLS and directly add an MLP classifier on the output of the encoder (node memory). In the second ablation experiment, we remove the Latent Generator from the INR2JLS framework, and employ an MLP decoder to directly map the node memory obtained from the DNG-Encoder to images for the reconstruction task. We then utilize another MLP to classify the node memory generated by the pretrained DNG-Encoder. Table 5 (Bottom) shows the experimental results. Our INR2JLS significantly outperforms its two variants with component removal, further demonstrating the necessity of the proposed Decoder and Latent Generator.

**Efficiency analysis.** Table 6 compares running time, memory usage, and computational complexity for processing a single INR. INR2JLS is significantly faster than other methods, with slightly higher memory usage than NG-GNN but much lower than NFN and NFT, and has the lowest computational complexity overall. We believe these advantages make INR2JLS a efficient choice for processing neural network weights.

Table 6: Inference efficiency comparison for INR classification on MNIST INR dataset. Results are based on the inference of a single INR by each method.

|  | NFN | NFT | NG-GNN | NG-T | INR2JLS (Ours) |
|---|---|---|---|---|---|
| Running Time (s) | $0.0082_{\pm 0.00009}$ | $0.0527_{\pm 0.00170}$ | $0.0124_{\pm 0.00070}$ | $0.0092_{\pm 0.00041}$ | $\mathbf{0.0047}_{\pm 0.00018}$ |
| Memory (MB) | 273.08 | 241.15 | $\mathbf{27.40}$ | 29.77 | 29.17 |
| Comp. Cost (GFLOPs) | 2.58 | 10.60 | 2.13 | 14.82 | $\mathbf{1.31}$ |

## 8 CONCLUSION

In this paper, we have introduced a novel method to model neural network weights as dynamic graphs. To process dynamic neural graphs, we propose Dynamic Neural Graph Encoder (DNG-Encoder) to handle the temporal dynamics intrinsic to neural network inference, maintaining the sequential flow of data through the layers. We further enhance our model's utility with INR2JLS, which maps INR weights into a joint latent space, providing a more informative and robust representation for downstream tasks. Extensive experiments demonstrate the effectiveness of our method.

**Limitation and Future Work.** Despite the significant improvements on classifying INRs using our method, its performance still lags behind that of CNNs on analogous image-space tasks. Developing a more powerful variant of temporal GNNs could potentially lead to further improvements. Moreover, since our experiments were primarily conducted on 2D images, extending the proposed approach to handle neural radiance fields would significantly broaden its potential applications.

## REPRODUCIBILITY

To ensure the reproducibility of our work, we have made all necessary resources and documentation available in both the main paper and supplementary materials. The full details of our model architecture, training setup, and experimental protocols are outlined in Section 6 and Appendix H. The source code for our proposed method, along with scripts to reproduce all experiments, has been anonymized and made available in the supplementary materials.

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

## A    RELATED WORK

**Implicit Neural Representations.** Recent works employ a neural network as a continuous function to implicitly represent the objects or shapes (Park et al. (2019)Genova et al. (2019a)Genova et al. (2019b)Michalkiewicz et al. (2019)Gropp et al. (2020)). This function takes an input (often a point coordinate) and outputs the corresponding feature value or property. SIREN (Sitzmann et al. (2020)) is a continuous implicit neural representation that utilizes sine as a periodic activation function. It excels at fitting complex signals, including natural images and 3D shapes. In our experiments, the INRs in the datasets we used with the form of SIRENs.

**Dynamic Graphs.** Dynamic graphs can be categorized into two main types: continuous-time dynamic graphs (CTDGs) and discrete-time dynamic graphs (DTDGs). A DTDG represents snapshots of a dynamic graph captured at regular time intervals, where each snapshot represents the graph structure at a specific timestamp and can be treated as a static graph. A CTDG can be represented by an initial state of a dynamic graph (essentially a static graph) and a sequence of events occurring at different timestamps. To address the complex structure and temporal information in dynamic graphs, substantial research has focused on the dynamic graph neural network. A common approach involves employing an encoder-decoder structure (Kazemi et al. (2020)). Here, the encoder is responsible for learning node embeddings, while the decoder utilizes these embeddings to perform downstream tasks. There has been a lot of work to handle DTDG and CTDG using neural networks, and one line of notable approach is leveraging recurrent neural networks (RNNs) to capture temporal dependencies and dynamics within the graph structure, such as methods for processing DTDG (Seo et al. (2018)Narayan & Roe (2018)Chen & Wang (2018)Chen et al. (2022), and methods for processing CTDG Rossi et al.Kumar et al. (2018)Kumar et al. (2019)Trivedi et al. (2017)Trivedi et al. (2019)).

**Learning in Deep Weight Spaces.** The intricate and high-dimensional nature of weight spaces in implicit neural representations (INRs) presents substantial challenges for extracting meaningful information about the encoded data. To tackle these challenges, some pioneers have focused on narrowing the effective weight space through constrained training processes (Bauer et al. (2023); Dupont et al. (2021); De Luigi et al. (2023)). However, Navon et al. (2023); Zhou et al. (2024a) argue that these methods overlook the permutation symmetry property of neural network weights. Ignoring this symmetry can expand the search space for optimal parameters, leading to decreased generalization and performance. To address this issue, they introduced permutation equivariant weight-space models. To further improve performance, Zhou et al. (2024b) recently introduced a transformer structure to address the problem. However, these methods require manual adaptation, which can be a burden to developers. To facilitate the processing of heterogeneous architectures, Kofinas et al. (2024) proposed modeling neural network weights as graphs, linking parameters similarly to a computation graph. Similarily, Graph Metanetworks (GMNs) Lim et al. (2024) also leverages graph neural networks to

process neural network weights as input, offering a generalizable, expressive, and symmetry-aware solution for diverse architectures, including multi-head attention, normalization layers, ResNet blocks, and group-equivariant layers. These approach, though innovative, mainly employs static graphs. In this paper, we suggest considering the temporal nature of neural networks' inference. We propose converting neural networks into dynamic graphs and introducing a temporal graph neural network to handle them.

# B   DYNAMIC NEURAL GRAPH SYMMETRY

## B.1   BACKGROUND AND DEFINITIONS

### B.1.1   NEURON PERMUTATION SYMMETRY IN NEURAL NETWORKS

Consider an $L$-layer Multilayer Perceptron (MLP) $\mathbf{M}$ with weight matrices $\{\mathbf{W}^l\}_{l=1}^L$ and biases $\{\mathbf{b}^l\}_{l=1}^L$. The network computes activations as:

$$\mathbf{h}^l = \sigma\left(\mathbf{W}^l \mathbf{h}^{l-1} + \mathbf{b}^l\right), \quad \text{for } l = 1, 2, \ldots, L, \tag{10}$$

where $\mathbf{h}^0$ is the input and $\sigma$ is an activation function.

**Neuron Permutation Symmetry:** Permuting the neurons within a hidden layer $l$ and appropriately adjusting the corresponding rows and columns of the weight matrices and biases leaves the function represented by the network unchanged. Formally, for any permutation $\pi^l$ of the neurons in layer $l$, there exists a transformed network $\tilde{\mathbf{M}}$ such that:

$$\tilde{\mathbf{W}}^l = \mathbf{P}^{\pi^l} \mathbf{W}^l \left(\mathbf{P}^{\pi^{l-1}}\right)^\top, \quad \tilde{\mathbf{b}}^l = \mathbf{P}^{\pi^l} \mathbf{b}^l, \tag{11}$$

where $\mathbf{P}^{\pi^l}$ is the permutation matrix corresponding to $\pi^l$.

## B.2   OBJECTIVE

Our goal is to prove that the dynamic neural graph $\mathcal{G}_T$ is equivariant to neuron permutations in the MLP $\mathbf{M}$. That is, permuting the neurons in any layer $l$ corresponds to permuting the nodes $\mathbf{v}^l$ in $\mathcal{G}_T$, and the graph update operations $\mathbf{O}_{t^l}$ are consistent under such permutations.

## B.3   PROOF OF EQUIVARIANCE

### B.3.1   DEFINING PERMUTATIONS IN NEURAL NETWORKS AND GRAPHS

Let $\pi^l$ be a permutation of the neurons in layer $l$ of the MLP, and let $\mathbf{P}^{\pi^l}$ be the corresponding permutation matrix. The permutation acts on the weights and biases as in Equation 11.

In the dynamic neural graph, the permutation $\pi^l$ acts on the nodes $\mathbf{v}^l$ and edges $\mathbf{e}^l$ as follows:

- **Nodes:** The permuted nodes are $\tilde{\mathbf{v}}^l = \mathbf{P}^{\pi^l} \mathbf{v}^l$.

- **Edges:** Each edge from node $i$ in $\mathbf{v}^{l-1}$ to node $j$ in $\mathbf{v}^l$ becomes an edge from node $\pi^{l-1}(i)$ to node $\pi^l(j)$ after permutation.

### B.3.2   NODE PERMUTATIONS CORRESPOND TO NEURON PERMUTATIONS

Since each node $\mathbf{v}_i^l$ corresponds to neuron $i$ in layer $l$ of the MLP, permuting the neurons directly corresponds to permuting the nodes:

$$\tilde{\mathbf{v}}_i^l = \mathbf{v}_{\pi^l(i)}^l. \tag{12}$$

### B.3.3 EDGE ADJUSTMENTS UNDER PERMUTATION

Edges $\mathbf{e}^l$ represent the connections (weights) between nodes in $\mathbf{v}^{l-1}$ and $\mathbf{v}^l$. The adjacency matrix $\mathbf{A}^l$ corresponding to edges $\mathbf{e}^l$ is related to the weight matrix $\mathbf{W}^l$.

Under permutations $\pi^{l-1}$ and $\pi^l$, the adjacency matrix transforms as:

$$\tilde{\mathbf{A}}^l = \mathbf{P}^{\pi^l} \mathbf{A}^l \left( \mathbf{P}^{\pi^{l-1}} \right)^{\top}. \tag{13}$$

This ensures that the structure of the graph remains consistent with the permuted network.

### B.3.4 EQUIVARIANCE OF GRAPH UPDATE OPERATIONS

The graph update operations $\mathbf{O}_{t^l}$ are defined independently of node identities and depend only on the layer structure. Therefore, they are consistent under permutations:

$$\tilde{\mathbf{O}}_{t^l} = \mathbf{O}_{t^l}. \tag{14}$$

Applying permutation $\pi^l$ to $\mathcal{G}_{t^l}$ after the graph update operations yields:

$$\tilde{\mathcal{G}}_{t^l} = \pi^l \left( \mathcal{G}_{t^l} \right). \tag{15}$$

### B.3.5 INDUCTIVE PROOF OVER LAYERS

We use mathematical induction over the layers $l$ to show that the graph remains equivariant under permutations.

**Base Case ($l = 1$)**   At $t^1$:

- The graph $\mathcal{G}_{t^1}$ consists of input nodes $\mathbf{v}^0$ and nodes $\mathbf{v}^1$.
- Permuting $\mathbf{v}^1$ corresponds to permuting neurons in layer 1.
- The update operations $\mathbf{O}_{t^1}$ are equivariant under $\pi^1$.

**Inductive Step**   Assume $\mathcal{G}_{t^{l-1}}$ is equivariant under permutations up to layer $l-1$. At $t^l$:

- Applying $\mathbf{O}_{t^l}$ to $\mathcal{G}_{t^{l-1}}$ adds nodes $\mathbf{v}^l$ and edges $\mathbf{e}^l$.
- Under permutation $\pi^l$, nodes and edges are permuted as per Equations 12 and 13.
- Thus, $\mathcal{G}_{t^l}$ remains equivariant under the combined permutations $\pi^{l-1}$ and $\pi^l$.

By induction, $\mathcal{G}_T$ is equivariant under neuron permutations at each layer.

## C EQUIVARIANCE OF THE DNG-ENCODER ON DYNAMIC GRAPHS

In this section, we prove that our proposed DNG-Encoder, when applied to dynamic graphs, is *equivariant* under node permutations. This property ensures that if the nodes of the input graph are permuted, the output will be permuted in the same way, maintaining consistency regardless of the node ordering.

### C.1 DEFINITION OF EQUIVARIANCE

A function $F$ operating on graphs is said to be **equivariant** to node permutations if, for any permutation $\pi$ and input graph $\mathcal{G}$, the following holds:

$$F(\pi \cdot \mathcal{G}) = \pi \cdot F(\mathcal{G}), \tag{16}$$

where $\pi \cdot \mathcal{G}$ denotes the graph obtained by permuting the nodes of $\mathcal{G}$ according to $\pi$, and similarly for $\pi \cdot F(\mathcal{G})$.

## C.2 EQUIVARIANCE OF THE MESSAGE PASSING FUNCTION

Our message passing function is defined differently for single-edge and multi-edge cases.

**Single-Edge Case** For the case where there is only one edge between a pair of nodes (i.e., for the dynamic neural graph converted from an MLP), the message function is defined in Equation 4 as:

$$\mathbf{m}_i(t^l) = \phi_m^{t^l}(\mathbf{s}_j(t^{l-}), \mathbf{e}_{ij}(t^l)) = \sum_{j \in \mathcal{N}_i} W_{m1}^{t^l} \mathbf{e}_{ij}(t^l) \odot W_{m2}^{t^l} \mathbf{s}_j(t^{l-}), \tag{17}$$

**Proof of Equivariance:**

Let $\pi$ be a permutation of the node indices. Under permutation $\pi$:

- Node $i$ becomes $\pi(i)$.
- The neighbor set $\mathcal{N}_i$ becomes $\mathcal{N}_{\pi(i)} = \{\pi(j) \mid j \in \mathcal{N}_i\}$.
- The edge from $j$ to $i$ becomes the edge from $\pi(j)$ to $\pi(i)$.
- Node states and edge features are permuted accordingly:

$$\mathbf{s}'_{\pi(j)}(t^{l-}) = \mathbf{s}_j(t^{l-}), \quad \mathbf{e}'_{\pi(j)\pi(i)}(t^l) = \mathbf{e}_{ji}(t^l). \tag{18}$$

The message for node $\pi(i)$ after permutation is:

$$\begin{aligned}
\mathbf{m}'_{\pi(i)}(t^l) &= \sum_{k \in \mathcal{N}_{\pi(i)}} W_{m1}^{t^l} \mathbf{e}'_{k\pi(i)}(t^l) \odot W_{m2}^{t^l} \mathbf{s}'_k(t^{l-}) \\
&= \sum_{k=\pi(j)} W_{m1}^{t^l} \mathbf{e}'_{\pi(j)\pi(i)}(t^l) \odot W_{m2}^{t^l} \mathbf{s}'_{\pi(j)}(t^{l-}) \\
&= \sum_{j \in \mathcal{N}_i} W_{m1}^{t^l} \mathbf{e}_{ji}(t^l) \odot W_{m2}^{t^l} \mathbf{s}_j(t^{l-}) \\
&= \mathbf{m}_i(t^l).
\end{aligned} \tag{19}$$

Therefore, we have:

$$\mathbf{m}'_{\pi(i)}(t^l) = \mathbf{m}_i(t^l), \tag{20}$$

which shows that the message passing function is equivariant under node permutations in the single-edge case.

**Multi-Edge Case** For the case where there are multiple edges between a pair of nodes (i.e., for the dynamic neural graph converted from a CNN), the message function is:

$$\mathbf{m}_i(t^l) = \sum_{j \in \mathcal{N}_i} \phi_h^{t^l} \left( \text{Concat} \left( \text{head}_{ij}^1(t^l), \dots, \text{head}_{ij}^N(t^l) \right) \right), \tag{21}$$

where each head is defined as:

$$\text{head}_{ij}^n(t^l) = W_{m1}^{t^l} \mathbf{e}_{ij,n}(t^l) \odot W_n^{t^l} \mathbf{s}_j(t^{l-}). \tag{22}$$

**Proof of Equivariance:**

Under permutation $\pi$, similar reasoning applies:

- Edge features are permuted: $\mathbf{e}'_{\pi(j)\pi(i),n}(t^l) = \mathbf{e}_{ji,n}(t^l)$.
- Node states are permuted: $\mathbf{s}'_{\pi(j)}(t^{l-}) = \mathbf{s}_j(t^{l-})$.

The message for node $\pi(i)$ is:

$$
\begin{aligned}
\mathbf{m}'_{\pi(i)}(t^l) &= \sum_{k \in \mathcal{N}_{\pi(i)}} \phi_h^{t^l} \left( \text{Concat} \left( \text{head}^1_{\pi(i)k}(t^l), \dots, \text{head}^N_{\pi(i)k}(t^l) \right) \right) \\
&= \sum_{k=\pi(j)} \phi_h^{t^l} \left( \text{Concat} \left( \text{head}^1_{\pi(i)\pi(j)}(t^l), \dots, \text{head}^N_{\pi(i)\pi(j)}(t^l) \right) \right) \\
&= \sum_{j \in \mathcal{N}_i} \phi_h^{t^l} \left( \text{Concat} \left( \text{head}^1_{ij}(t^l), \dots, \text{head}^N_{ij}(t^l) \right) \right) \\
&= \mathbf{m}_i(t^l).
\end{aligned} \tag{23}
$$

Thus, the message passing function remains equivariant under node permutations in the multi-edge case as well.

### C.3 EQUIVARIANCE OF THE RECURRENT MEMORY UPDATING

The recurrent memory update function is defined as:

$$
\mathbf{s}_i(t^l) = \phi_u^{t^l}(\mathbf{m}_i(t^l), \mathbf{v}_i(t^l)) = \text{GRU}(\mathbf{m}_i(t^l), \mathbf{v}_i(t^l)), \tag{24}
$$

where $\mathbf{v}_i(t^l)$ is the feature of node $i$ at time $t^l$.

Under permutation $\pi$, node features are permuted:

$$
\mathbf{v}'_{\pi(i)}(t^l) = \mathbf{v}_i(t^l). \tag{25}
$$

Messages are permuted:

$$
\mathbf{m}'_{\pi(i)}(t^l) = \mathbf{m}_i(t^l). \tag{26}
$$

Therefore, the updated memory state for node $\pi(i)$ is:

$$
\begin{aligned}
\mathbf{s}'_{\pi(i)}(t^l) &= \phi_u^{t^l}(\mathbf{m}'_{\pi(i)}(t^l), \mathbf{v}'_{\pi(i)}(t^l)) \\
&= \text{GRU}(\mathbf{m}_i(t^l), \mathbf{v}_i(t^l)) \\
&= \mathbf{s}_i(t^l).
\end{aligned} \tag{27}
$$

This shows that the memory update function is equivariant under node permutations.

### C.4 EQUIVARIANCE OF GRAPH UPDATE EVENTS

Our dynamic graph evolves through graph update events defined at each time $t^l$, as specified in Equation 3. These events include node addition $(+V)$, edge addition $(+E)$, node deletion $(-V)$, and edge deletion $(-E)$.

Under permutation $\pi$:

- Added nodes $\mathbf{v}^l$ become $\mathbf{v}^{l\prime} = \{\pi(i) \mid \mathbf{v}_i^l \in \mathbf{v}^l\}$.
- Added edges $\mathbf{e}^l$ become $\mathbf{e}^{l\prime} = \{(\pi(i), \pi(j)) \mid (\mathbf{v}_i^{l-1}, \mathbf{v}_j^l) \in \mathbf{e}^l\}$.
- Deleted nodes and edges are permuted similarly.

Since the graph update operations are applied consistently to the permuted nodes and edges, the sequence of graph updates remains equivariant under node permutations.

### C.5 CONCLUSION

By demonstrating that each component of our DGN-Encoder, we establish that the entire model maintains equivariance when applied to dynamic graphs. This property ensures that the model's outputs are consistent regardless of the node ordering, capturing the intrinsic structure of the graph without being influenced by arbitrary node labels.

## D  How Does the DNG-Encoder Exhibit the Expressiveness of Neural Networks?

Using the DNG-Encoder to update nodes in dynamic neural graph ideally simulates the sequential updating pattern of neural networks. This approach effectively avoids the inverse problem typically encountered in static neural graphs as discussed in Section 2.3, so as can better exhibit the expressiveness of neural networks. Below, we discuss how the DNG-Encoder model better represents the expressiveness of MLPs compared to the static neural graph-based model.

An MLP $\mathbf{M}$ can be transformed into a dynamic neural graph $\mathcal{G}_T$ or a static neural graph $\mathcal{G}_s$. In $\mathcal{G}_T$, we update node representations asynchronously at each layer by modifying the graph structure at each timestamp to align with the forward pass process of $\mathbf{M}$. The graph structure under each timestamp of $\mathcal{G}_T$ is a layer-by-layer snapshot taken from $\mathcal{G}_s$. It contains the neurons involved in the computation of the neural network in each forward pass step and simulates the topology of these neurons. For example, the graph state $\mathcal{G}_{t^l}$ ($0 < l \leq L$) of $\mathcal{G}_T$ only contains nodes at the $l$-th layer and the $(l-1)$-th layer of $\mathcal{G}_s$ and edges at the $l$-th layer of $\mathcal{G}_s$. The initial representations of all nodes and edges present in $\mathcal{G}_T$ over the time span $T = [t^0 : t^L]$ can be defined as $\{\mathbf{v}^0(t^0), \mathbf{v}^1(t^1-), ..., \mathbf{v}^L(t^L-)\}$ and $\{\mathbf{e}^1(t^1-), ..., \mathbf{e}^L(t^L-)\}$, where $t^l-$ denotes the timestamp immediately before $t^l$. Similar to $\mathcal{G}_s$, these nodes and edges represent the biases and weights of each layer in $\mathbf{M}$. According to DNG-Encoder defined in Section 4.1, a node representation $\mathbf{v}_i^l(t^l)$ at timestamp $t^l$ can be obtained by using the following equation:

$$\mathbf{v}_i^l(t^l) = \phi_u(\mathbf{v}_i^l(t^l-), \sum_{j \in N_i} \phi_m^{t^l} \left( \mathbf{e}_{ij}^l \left( t^l- \right), \mathbf{v}_j^{l-1} \left( t^l- \right) \right), \tag{28}$$

where $\phi_m^{t^l}$ is the message function at $t^l$, corresponding to the Message Function in the DNG-Encoder. $\phi_u$ is the node update function shared for all timestamps, corresponding to the GRU module in DNG-Encoder. For example, at the first timestamp $t^1$, $\mathbf{e}_{ij}^1(t^1-)$ is an initial representation of an edge newly added at $t^1$, corresponding to $W_{ij}^1$ in Equation 1, while $\mathbf{v}_j^0(t^1-)$ corresponds to the input $\mathbf{x}_j$ in Equation 1. From Equation 4, we know that $\phi_m^{t^1}$ can approximates the multiplication operation between two given inputs. Thus, given $\mathbf{e}_{ij}^1(t^1-)$ and $\mathbf{v}_j^0(t^1-)$, its output can represent $W_{ij}^1 \mathbf{x}_j$ in Equation 1. $\mathbf{v}_i^1(t^1-)$ is the initial representation of a newly added node at $t^1$, corresponding to $b_i^1$ in Equation 1. From Equation 6, given the aggregation of the outputs of $\phi_m^{t^1}$ and $\mathbf{v}_i^1(t^1-)$, $\phi_u$ can easily approximate the computation of adding $b_i^1$ to $\sum_j W_{ij}^1 \mathbf{x}_j$ and then applying an activation function. In this way, we say that $\mathbf{v}_i^1(t^1)$ can directly represent $\mathbf{a}_i^1$.

According to the discussion in Section and 2.2 and 2.3, it is evident that the static neural graph-based model can also approximate the first forward pass step of $\mathbf{M}$ easily using one MPNN layer. However, it faces challenges in approximating subsequent forward pass steps due to the emergence of the "inverse problem". The following demonstrates how the DNG-Encoder avoids the inverse problem during computation, thereby enabling it to easily approximate the forward pass process for all steps of $\mathbf{M}$. We here define the process of $\mathbf{M}$ to obtain the $i$-th activation $a_i^l$ at the $l$-th layer as follows:

$$\mathbf{a}_i^l = \sigma(b_i^l + \sum_j W_{ij}^l \mathbf{a}_j^{l-1}). \tag{29}$$

At any timestamp $t^l$, $\mathbf{v}_i^l(t^l-)$ in Equation 28 is the initial representation of the newly added node, corresponding to $b_i^l$ in Equation 29. Smilarly, $\mathbf{e}_{ij}^l(t^l-)$ in Equation 28 is the initial representation of the newly added edge, corresponding to $W_{ij}^l$ in Equation 29. Besides, $\mathbf{v}_j^{l-1}(t^l-)$ in Equation 28 corresponds to $a_j^{l-1}$.

Following the expressivity in Kofinas et al. (2024), this suggests that the message functions $\phi_m^{t^l}, \ldots, \phi_m^{t^L}$, which have the same structure at all timestamps, along with the shared update function $\phi_u$, can accurately model all forward pass steps of $\mathbf{M}$. Since the inputs to the message/update functions are simple and do not contain extra complex terms, the approximation is straightforward, eliminating the risk of inverse problems. Therefore, the proposed DNG-Encoder can maximally approximate the forward pass of $\mathbf{M}$.

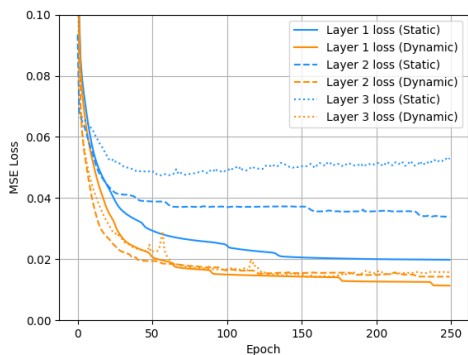

Figure 5: The Mean Squared Error (MSE) for fitting the activations of each layer of an MLP using the static neural graph-based model and the dynamic neural graph-based model, respectively.

# E    EXPERIMENT FOR COMPARING STATIC AND DYNAMIC NEURAL GRAPH

## E.1    ABILITY TO APPROXIMATE THE FORWARD PASS PROCESS.

The following experiment evaluates the capability of static and dynamic neural graph-based models to approximate the forward pass of an input neural network by comparing their performance in fitting activations across varying numbers of layers in MLPs.

To initiate the experiment, we randomly generate the weights and biases for 1000 three-layer MLPs as training data, followed by the generation of 500 MLPs with identical structures as testing data. Then, we randomly generate data for 1500 MLPs. We use the generated data as input for the generated MLPs and save the activation values for each network across different inputs.

Our objective is to employ an MPNN on both static and dynamic neural graph to generate node embeddings, fitting these embeddings with their corresponding activations. A better fit for activations means that the model better approximates the forward pass of the MLP. To highlight the impact of static and dynamic neural graph framework on fitting results, we employ identical message function $\phi_m$ and node update function $\phi_u$ as defined in Section 2.2 on both types of neural graphs. $\phi_m$ concatenates the source node feature and edge feature, then utilizing a two-layer MLP to generate a message. $\phi_u$ concatenates the aggregation of the messages with target node feature, then utilizing a two-layer MLP to generate the target node embedding. For the static neural graph, we adopt the method proposed by Kofinas et al. (2024), which employing an L-layer MPNN to simulate the forward pass process of an L-layer MLP. We also utilize their approach to update edges. For the dynamic neural graph, we use the method defined in Section 3.1 to construct the dynamic neural graph and use $\phi_m$ and $\phi_u$ to update the node embeddings layer by layer as the timestamps evolve.

Figure 5 illustrates the MSE loss of models based on two types of graphs fitting activations across three different number of layers on the test set. It is evident that the dynamic neural graph-based model consistently performs well in fitting the activations across all three layers. However, the performance of the static neural graph-based model closely matches that of the dynamic neural graph-based model only when fitting the activations of the first layer. In contrast, at the second layer, the fitting performance of the static model declines significantly compared to the dynamic model, with this discrepancy becoming more pronounced as layer depth increases.

The above observation aligns with our discussion in Section 2.3, indicating that the method proposed by Kofinas et al. (2024), which is based on static neural graphs, is primarily effective for simulating only the initial forward pass of the input neural network. However, it may fail to approximate the functionality of subsequent layers. In contrast, our proposed dynamic neural graph framework is capable of accurately simulating all forward pass steps of the input neural network.

Table 7: Test accuracy (%) for the INR classification task utilizing 10 views of input INRs as data augmentation across various datasets. #Params denotes the number of parameters required in the inference. We do not use probe features for NG-GNN and NG-T. Our DNG-Encoder consistently outperforms static graph-based classifiers across all three datasets.

|  | #Params | MNIST | FashionMNIST | CIFAR-10 |
|---|---|---|---|---|
| NFN | $\sim$135M | 92.9$\pm$0.38 | 75.6$\pm$1.07 | 46.6$\pm$0.13 |
| NG-GNN | $\sim$0.3M | 79.6$\pm$1.3 | 71.1$\pm$0.42 | 43.94$\pm$0.06 |
| NG-T | $\sim$0.4M | 83.43$\pm$0.12 | 72.13$\pm$0.51 | 44.69$\pm$0.03 |
| DNG-Encoder(ours) | $\sim$0.4M | **96.6**$\pm$0.04 | **78.4**$\pm$0.61 | **54.0**$\pm$0.07 |

### E.2    COMPARISON ON CLASSIFYING INRS

The following experiment presents the performance of two graph frameworks, dynamic graph and static graph, on the INR classification task. To emphasize the impact of the graph framework on classification results, we use only the DNG-Encoder from INR2JLS framework as the dynamic graph-based classifier. For the static graph-based classifiers, we employ NG-GNN and NG-T Kofinas et al. (2024) without utilizing probe features. Probe features, which are intermediate outputs generated by different inputs during the forward pass of neural networks, are excluded from this analysis. This is because probe features accurately capture the forward pass process of neural networks, directly representing the expressiveness of the neural networks. As a result, they may potentially prevent us from observing the ability of static graphs themselves to exhibit the expressiveness of the neural networks. Table 7 presents the classification performance of the three models on the MNIST INRs dataset, FashionMNIST INRs dataset and CIFAR-10 INRs dataset. It is evident that the DNG-Encoder significantly outperforms both NG-GNN and NG-T, indicating that dynamic graphs can better exhibit the expressiveness of neural networks compared to static graphs.

## F    ANALYSIS ON COMPUTATIONAL COST

Since our method does not require a decoder for inference, we focus this analysis on providing theoretical results for the computational costs during the inference within the encoder.

To determine the time complexity of applying an $L$-layer Message Passing Neural Network (MPNN) on a graph, we consider the following characteristics:

- **Number of nodes per MLP layer:** $n$. For simplicity, we assume each MLP layer has $n$ neurons.

- **Dimension of node/edge features:** $d$. For simplicity, we assume the dimensions of edge and node features are the same.

- **Number of MLP layers:** $L$.

### F.1    TIME COMPLEXITY OF OUR METHOD

1. **Message Computation (Equation 4):**
   - Per edge computation: $O(d^2)$ (including edge feature transformation).
   - Total edges computation: $O(n^2 \cdot L \cdot d^2)$. There are $n^2$ edges in an MLP layer, and $L$ total layers.

2. **Aggregation (Equation 4):**
   - Per node aggregation: $O(n \cdot d)$. Each node aggregates messages from its $n$ neighbors in the graph from the previous time step.
   - Total nodes: $O(n^2 \cdot L \cdot d)$.

3. **Recurrent Memory Updates (Equation 6):** $O(n \cdot L \cdot d^2)$.

**Total Computational Complexity:** $O(n^2 \cdot L \cdot d^2 + n^2 \cdot L \cdot d + n \cdot L \cdot d^2)$

**Analysis:** For large input networks, the term $O(n^2 \cdot L \cdot d^2)$ dominates the overall computational cost.

### F.2 SPACE COMPLEXITY OF OUR METHOD

1. **Node and Edge Features:** $O(n^2 \cdot d + n \cdot d)$. We store the node and edge features of the graph from the previous time step, corresponding to the previous MLP layer.

2. **Memory (Equation 6):** $O(n \cdot d)$.

**Total Space Complexity:** $O(n^2 \cdot d + n \cdot d)$

## G NEURAL NETWORKS AS DYNAMIC NEURAL GRAPHS

### G.1 ADDITIONAL COMPONENTS IN CNNS AS DYNAMIC NEURAL GRAPHS

In Section 3.2, we outlined how to convert the fundamental modules of CNNs - the convolutional layers and the linear layers - into modules within the dynamic neural graph. In modern CNNs, besides convolutional and linear layers, there are often additional components like the flattening layer and residual connections (He et al. (2016)). To ensure our dynamic neural graph framework is applicable to a wide range of CNN architectures, we define how to convert flatten layers and residual connections to the modules in dynamic neural graphs.

**Flattening layer.** The flattening layer in CNNs is used to convert multi-dimensional feature maps into a single feature vector that can be accepted by fully connected layers for subsequent operations. The dimensions of the feature map output by a convolutional layer at the $l$-th layer of a CNN are $h_{ft}^l \times w_{ft}^l \times c^l$, where $h_{ft}^l$ and $w_{ft}^l$ represent the spatial dimensions of the feature map, $c^l$ denotes the number of channels of the feature map. Assuming we flatten the feature map to a single feature vector of dimension $d^l$, where $d^l = h_{ft}^l \times w_{ft}^l \times c^l$. A linear layer is then utilized at the $(l+1)$-th layer of the CNN to perform a linear transformation on this flattened feature vector, yielding a vector of dimension $d^{l+1}$.

In the dynamic neural graph, $c^l$ corresponds to the number of nodes in $\mathbf{v}^l$, and $d^{l+1}$ corresponds to the number of nodes in $\mathbf{v}^{l+1}$. We can conceptualize the function of the flattening layer as generating $h_{ft}^l \times w_{ft}^l$ *virtual vertices* within each node in $\mathbf{v}^l$, and they are subsequently connected to nodes $\mathbf{v}^{l+1}$. Virtual vertices do not contain any feature information, they are only used to indicate the connection relationship between their carriers $\mathbf{v}^l$ and nodes $\mathbf{v}^{l+1}$. Each connection between a virtual vertex and a node in $\mathbf{v}^{l+1}$ corresponds to a single weight scalar in the weight matrix $W^{l+1}$ of the linear layer at the $(l+1)$-th layer. In essence, this implies that each node in $\mathbf{v}^l$ is linked to a node in $\mathbf{v}^{l+1}$ via $h_{ft}^l \times w_{ft}^l$ edges by utilizing virtual vertices embedded within itself. In addition, to maintain consistency in the number of edges between each pair of nodes throughout the entire CNN, we employ the same method as proposed in Section 3.2 to pad the edges between $\mathbf{v}^l$ and $\mathbf{v}^{l+1}$. To provide a more intuitive understanding of the process of converting the flattening layer to the dynamic neural graph, we present an example in Figure 6.

**Residual Connections.** Residual connections are used in neural networks to address the gradient vanishing problem. Specifically, a residual connection in a neural network allows the input to bypass one or more layers and be added directly to the output. If a residual connection is established between the output of the $l$-th layer and the output of the $(l+r)$-th layer in a CNN, then within the corresponding dynamic neural graph, we define events occurring at timestamp $t^{l+r}$ as the addition of nodes $\mathbf{v}^l$ at the $l$-th layer, in addition to the addition and deletion of nodes and edges as defined in Section 3.2. Additionally, we add new edges to ensure that each node $\mathbf{v}_i^l$ at the $l$-th layer is connected to node $\mathbf{v}_i^{l+r}$ at the $(l+r)$-th layer via a single edge. Considering the potential differences brought by residual connections with different time spans or layer spans to the update of target nodes, we define the edge feature of each of these edges between $\mathbf{v}^l$ and $\mathbf{v}^{l+r}$ as $\mathbf{e}_{res,i}^{l+r} = \theta_{res}r$, where $\theta_{res} \in \mathbb{R}^{d_e}$ is a learnable vector.

### G.2 TRANSFORMERS AS DYNAMIC NEURAL GRAPHS

In the Transformer, the core modules are multi-head self-attention layers. Assuming there are $h$ heads in a multi-head self-attention layer. For each head $head_i$, the input $\mathbf{X}$ with a dimension of $d_{\text{model}}$ [3] is

---

[3]In general, $\mathbf{X} \in \mathbb{R}^{L \times d_{\text{model}}}$. We omit the sequence length $L$ here for clarity.

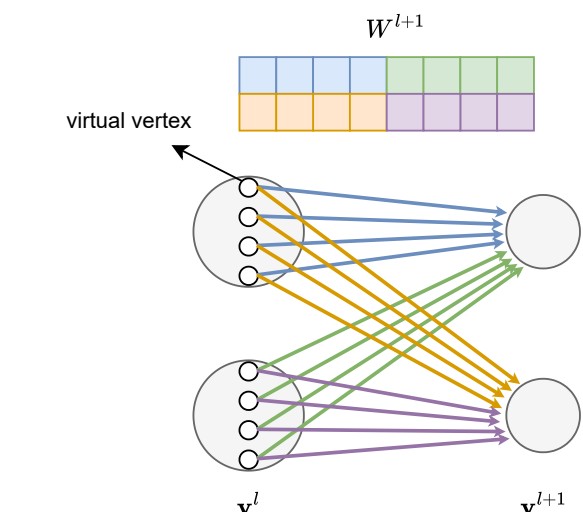

Figure 6: An example for converting the flattening layer to the dynamic neural graph, where edges correspond to the weight scalar of the same color.

firstly transformed into $\mathbf{Q}_i \in \mathbb{R}^{d_k}$, $\mathbf{K}_i \in \mathbb{R}^{d_k}$ and $\mathbf{V}_i \in \mathbb{R}^{d_v}$ through three linear projections.

$$\mathbf{Q}_i = \mathbf{X}\mathbf{W}_i^Q, \tag{30}$$

$$\mathbf{K}_i = \mathbf{X}\mathbf{W}_i^K, \tag{31}$$

$$\mathbf{V}_i = \mathbf{X}\mathbf{W}_i^V, \tag{32}$$

where $\mathbf{W}_i^Q \in \mathbb{R}^{d_{\text{model}} \times d_k}$, $\mathbf{W}_i^K \in \mathbb{R}^{d_{\text{model}} \times d_k}$ and $\mathbf{W}_i^V \in \mathbb{R}^{d_{\text{model}} \times d_v}$. The scaled dot-product attention is then computed using $\mathbf{Q}_i$, $\mathbf{K}_i$ and $\mathbf{V}_i$, producing $\mathbf{Z}_i \in \mathbb{R}^{d_v}$.

$$\mathbf{Z}_i = \text{Attention}(\mathbf{Q}_i, \mathbf{K}_i, \mathbf{V}_i) = \text{Softmax}\left(\frac{\mathbf{Q}_i\mathbf{K}_i^\top}{\sqrt{d_k}}\right)\mathbf{V}_i \tag{33}$$

Finally, the $\mathbf{Z}_i$ from all heads are concatenated, and the output $\mathbf{Y}$ with a dimension of $d_{\text{model}}$ is produced through a linear transformation $\mathbf{W}^O \in \mathbb{R}^{hd_v \times d_{\text{model}}}$.

To convert a multi-head self-attention layer into a dynamic neural graph while still simulating its forward pass process, we divide the multi-head self-attention layer into three timestamps within the dynamic neural graph.

In the first timestamp, we simulate the linear transformation from $\mathbf{X}$ to $\mathbf{Q}_i$, $\mathbf{K}_i$ and $\mathbf{V}_i$. We begin by adding $d_{\text{model}}$ nodes to the graph, with each node representing a dimension of the input $\mathbf{X}$. Next, for each head, we add $d_k + d_k + d_v$ nodes initialized as zero vectors, corresponding to the dimensions of $\mathbf{Q}_i$, $\mathbf{K}_i$ and $\mathbf{V}_i$. Additionally, we add each element of the weight matrices $\mathbf{W}_i^Q$, $\mathbf{W}_i^K$ and $\mathbf{W}_i^V$ as a single edge to the graph, connecting the corresponding nodes. Thus, for all heads, we add $h \times (d_k + d_k + d_v)$ nodes and $h \times (d_{\text{model}} \times d_k + d_{\text{model}} \times d_k + d_{\text{model}} \times d_v)$ edges.

In the second timestamp, we simulate the computation process of scaled dot-product attention. First, we delete the nodes corresponding to the input and the edges corresponding to $\mathbf{W}^Q$, $\mathbf{W}^K$ and $\mathbf{W}^V$, keeping only the nodes representing $\mathbf{Q}$, $\mathbf{K}$ and $\mathbf{V}$. It can be observed from Equation 33 that the dot product of $\mathbf{Q}_i$ and $\mathbf{K}_i^\top$, the division by $\sqrt{d_k}$ and the element-wise multiplication with $\mathbf{V}_i$ are parameter-free operations. Inspired by the method proposed by Kofinas et al. (2024), we design a simple graph structure to fit these operations. Specifically, for each head, we augment the graph by adding $d_v$ nodes initialized with zero vectors, corresponding to the dimensions of $\mathbf{Z}_i$. Additionally, we add edges connecting each newly added node to the nodes corresponding to $\mathbf{Q}_i$, $\mathbf{K}_i$ and $\mathbf{V}_i$. These edges are defined as learnable vectors, allowing them to fit the parameter-free operations mentioned above during training. For all heads, we add a total of $h \times d_v$ nodes and $h \times (d_k \times d_v + d_k \times d_v + d_v \times d_v)$ edges in this timestamp.

In the last timestamp, we simulate the process of mapping the concatenation of $\{\mathbf{Z}_1, \mathbf{Z}_2, ..., \mathbf{Z}_h\}$ to the output $\mathbf{Y}$. The process of performing the linear transformation using $\mathbf{W}^O$ is the same as the linear transformation using a linear layer in an MLP. Therefore, we can use the same method employed to convert linear layers into dynamic neural graphs in MLPs to transform this linear transformation process.

# H    DETAILS OF EXPERIMENTAL SETUP

Below, we provide additional detailed explanations of the experiments outlined in Section 6.

## H.1    CLASSIFY INRS WITH INR2JLS

### H.1.1    DATASETS

We applied the INR2JLS framework to classify images from the open-source MNIST, Fashion MNIST, and CIFAR-10 datasets as proposed by Zhou et al. (2024a). The INRs in these datasets are structured as three-layer MLPs with a hidden dimension of 32, utilizing the sine function as the activation function. These MLPs, employing the sine activation function, are commonly known as SIRENs (Sitzmann et al. (2020)). Following the strategy of splitting the datasets proposed by Zhou et al. (2024a), the datasets were split into 45,000 (MNIST, CIFAR) or 55,000 (FashionMNIST) training images, 5,000 validation images, and 10,000 test images. The training set is augmented by training 10 additional copies of SIRENs with different initializations for each training image, and each validation and test image has a single SIREN. Furthermore, we generate the CIFAR-100 INRs dataset following the methodology used by Zhou et al. (2024a) for generating the CIFAR-10 INRs dataset. Specifically, we train three-layer SIRENs with a hidden dimension of 32 for 5000 steps, employing the Adam optimizer with a learning rate of 5e-5. Additionally, we also train 10 additional copies of SIRENs with different initializations for each training image, while each image in the validation set and test set retains a single SIREN.

### H.1.2    MODELS

In the dynamic neural graph, the Fourier size of each node feature and edge feature is set to 128, and the Fourier scale is 3. In the DNG-Encoder of the INR2JLS framework, both the Message Function and the GRU have hidden dimensions of 512. In the Latent Generator, each spatial vector has a dimension of 512, and each output latent vector has a dimension of 128.

When no image augmentation is applied, we use 49 spatial vectors to generate 49 latent vectors for the MNIST and Fashion MNIST datasets. These latent vectors are then reshaped into a feature map with dimensions $7 \times 7 \times 128$. For the CIFAR-10 and CIFAR-100 datasets, 64 spatial vectors are used to generate 64 latent vectors, which are reshaped into a feature map with dimensions $8 \times 8 \times 128$. With image augmentation, which includes rotation and flipping, we employ $49 \times 6 = 294$ spatial vectors to generate 294 latent vectors for the MNIST and Fashion MNIST datasets. These vectors are reshaped into a feature map with dimensions $7 \times 7 \times (128 \times 6) = 7 \times 7 \times 768$. For the CIFAR-10 and CIFAR-100 datasets, we use $64 \times 6 = 384$ spatial vectors to generate 384 latent vectors, which are reshaped into a feature map with dimensions $8 \times 8 \times (128 \times 6) = 8 \times 8 \times 768$.

For the reconstruction task, we employ two transposed convolutional layers as a decoder, both layers with the kernel size of 4, the stride of 2, and the padding of 1 for all four datasets. The out channels of the first transposed convolutional layer are 256, the out channels of the second transposed convolutional layer are 1 (MNIST, Fashion MNIST) or 3 (CIFAR-10, CIFAR-100). The total number of parameters of the model is $5M$ for the MNIST and Fashion MNIST datasets and $6.1M$ for the CIFAR-10 and CIFAR-100 datasets.

For the classification task, we fix the trained DNG-Encoder and Latent Generator from the reconstruction task and use them for generating feature maps. For the MNIST and Fashion MNIST datasets, we utilize a classifier comprising two convolutional layers followed by a three-layer MLP. The classifier has the hidden dimension of 256, with the dropout rate of $0.5$ applied between each layer. For the CIFAR-10 and CIFAR-100 dataset, the classifier structure maintains the same as above, except for adjusting the dropout rate between convolutional layers to 0.2. The total parameters of the model is $5M$ for MNIST and Fashion MNIST, $6.7M$ for CIFAR-10 and CIFAR-100.

Table 8: A complete version of Table 2 by including training loss and standard deviation. Train and test MSE loss (lower is better) for MNIST erosion, MNIST dilation, MNIST gradient, and FashionMNIST gradient tasks with 10 views of input INRs as data augmentation. Our method outperforms the state-of-the-art on all tasks.

| | MNIST (erosion) | MNIST (dilation) | MNIST (gradient) | FashionMNIST (gradient) |
|---|---|---|---|---|
| NFN(HNP) | 0.0235±0.0010 | 0.0694±0.0007 | 0.0542 ±0.0003 | 0.0885±0.0006 |
| | 0.0217±0.0004 | 0.0628±0.0009 | 0.0541±0.0011 | 0.0843±0.0020 |
| NFN(NP) | 0.0221±0.0005 | 0.0582±0.0003 | 0.0526±0.0014 | 0.0920±0.0003 |
| | 0.0214±0.0007 | 0.0628±0.0001 | 0.0537±0.0006 | 0.0857±0.0001 |
| NFT | 0.0195±0.0004 | 0.0473±0.0006 | 0.0474±0.0005 | 0.0795±0.0009 |
| | 0.0194 ±0.0002 | 0.0510±0.0004 | 0.0484±0.0007 | 0.0800±0.0002 |
| NG-GNN | 0.0408±0.0005 | 0.0512±0.0003 | 0.0875±0.0002 | 0.0986±0.0002 |
| | 0.0417±0.0004 | 0.0547±0.0003 | 0.0907±0.0020 | 0.1002±0.0013 |
| NG-T | 0.0182±0.0002 | 0.0432±0.0007 | 0.0461±0.0008 | 0.0743±0.0007 |
| | 0.0193±0.0007 | 0.0486±0.0003 | 0.0484±0.0004 | 0.0777±0.0006 |
| DNG-Encoder(ours) | 0.0045±0.0005 | 0.0074±0.0006 | 0.0100 ±0.0013 | 0.0318±0.0011 |
| | **0.0071**±0.0004 | **0.0125**±0.0005 | **0.0153** ±0.0007 | **0.0434**±0.0015 |

### H.1.3 TRAINING

For the reconstruction task, we set the training batch size to 64 and use the Adam optimizer with a learning rate of 1e-4. The model is trained for 400,000 steps, and we apply an early stopping strategy that choosing the model showing the best performance on the validation set, as described by Zhou et al. (2024a).

For the classification task, the training batch size is set to 128, and we use the AdamW optimizer with a learning rate of 1e-4. The model is trained for 200,000 steps, and the early stopping strategy is also employed.

### H.2 EDITING INRS

We used the same MNIST INRs dataset and FashionMNIST INRs dataset as that used in the classification task. The training set of each dataset contains 10 views of INRs. Specifically, we applied three visual transformations to the images in these two datasets: dilation, erosion, and morphological gradient. Dilation indicates that the size of the object in the image is increased by adding pixels on the edge of the object, erosion indicates that the size of the object in the image is decreased by eliminating pixels on the edge of the object, morphologic gradient represents the edge of an object obtained by the difference between dilation and erosion. We did four experiments: MNIST Dalation, MNIST Erosion, MNIST Gradient and fashion MNIST Gadient.

We also use the INR2JLS framework to generate images that are visually transformed. Specifically, we use the same model and training strategy to process the input INRs as we did for the image reconstruction task, but change the reconstructed target from the original image to the visually transformed image. We do not use image augmentation. Table 8 shows the train loss and test loss of our model compared to other models on the same datasets. We can see that the both train loss and test loss of our model are much lower than other models. It is worth noting that the NFT model does not provide a complete training code, so I use the data provided in Zhou et al. (2024b), that is, the performance on the dataset with 1 view INRs in training set.

### H.3 PREDICTING CNN CLASSIFIER

We use DNG-Encoder to generate the memory of the nodes in the last layer at the last timestamp of the input CNNs, and then we map the memory of these nodes to a scalar value through an MLP head, representing the test accuracy of our prediction. For DNG-Encoder, we use a multi-head Message Function introduced in Section 5 to process the information that generates the dynamic neural graph converted from CNNs. In the model, the hidden dimensions of the Message Function and GRU are set to 128. The dimension of each head of the Message Function is 32. The MLP head is a three-layer

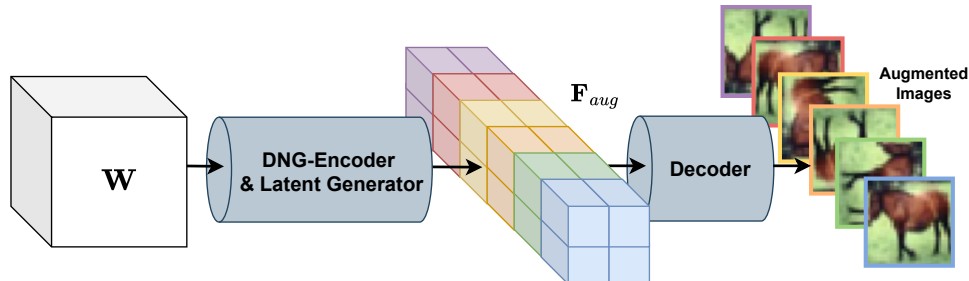

Figure 7: Data augmentation (rotation and flipping) for the INR2JLS framework.

Table 9: Test accuracy (%) for the INR classification task by using INR2JLS framework with different data augmentation strategies on MNIST, FashionMNIST, CIFAR-10 and CIFAR-100 INR datasets.

|  | MNIST | FashionMNIST | CIFAR-10 | CIFAR-100 |
|---|---|---|---|---|
| No Augmentation | 98.5±0.00 | 89.5±0.07 | 66.4±0.19 | 32.9±0.31 |
| Adding Noise Augmentation | 98.4±0.01 | 89.5±0.06 | 67.3±0.38 | 33.0±0.24 |
| Rotation&Flip | **98.6**±0.01 | **90.6**±0.07 | **73.2**±0.28 | **42.4**±0.32 |

MLP, and its hidden dimension is 1024. We set the training batch size to 128, use Adam as the optimizer with learning rate of 1e-4, train for 200 epochs, and use early stopping.

## H.4 DATA AUGMENTATION FOR INR2JLS

Data augmentation for images often introduces various image transformations during training. This allows the model to learn features and patterns under diverse conditions, enhancing its adaptability to input images and improving overall model generalization. In our proposed data augmentation method for the INR2JLS framework, we aim to enrich the feature map outputted by the DNG-encoder with a broader range of features and patterns present in the original image. This enhancement subsequently boosts the generalization ability of the classification model based on the feature map. Specifically, we rotate and flip the image corresponding to each INR, generating multiple images for each INR. Our model then learns a feature map $\mathbf{F}_{aug}$ with increased channels from the given INR to reconstruct multiple images. For a given input INR, we generate more latent vectors by defining more distinct spatial vectors to fuse with the same node memory $\mathbf{s}(t^L)$, and then permute these new latent vectors to generate a feature map $\mathbf{F}_{aug}$ with more channels. The rotation and flipping of each image in the dataset essentially alter the spatial arrangement of the pixels of the image, reflecting a same spatial arrangement of the input coordinates of each INR. After obtaining $\mathbf{s}(t^L)$ containing semantic information of the INR, we use different spatial vectors to simulate various permutations of input coordinates, and fuse them with $\mathbf{s}(t^L)$ in the latent space to decode the image. This process simulates the INR use different permutations of input coordinates to output images with different pixel arrangements. While the feature map $\mathbf{F}_{aug}$ can reconstruct distinct images to some extent, they inherently encapsulate a variety of features and patterns from these images. Consequently, the classifier can achieve better generalization when performing classification tasks based on $\mathbf{F}_{aug}$. Figure 7 shows how to employ data augmentation in the INR2JLS framework.

We also proposed a data augmentation method of adding noise to the image to compare with the above proposed method of rotating and flipping the image. For the method of rotating and flipping the image, we apply five transformations on the image, including clockwise rotations of 90, 180 and 270 degrees, as well as horizontal and vertical flips, which consistent with the settings in Section 6.1. For the method of adding noise, we add Gaussian noise with a mean of 0 and a standard deviation of 0.06 the original image. Table 9 shows the performance improvement of the two methods for the classification task. It can be found that adding noise has almost no significant improvement in the performance of the model, and even get worse performance on the MNIST dataset. The rotation and flip method can improve the performance for all datasets, especially for the more complex CIFAR-10 and CIFAR-100 datasets.

# I ADDITIONAL EMPIRICAL ANALYSIS

We also conduct additional analyses on the influence of positional encoding and non-linearity embeddings on the INR2JLS framework.

## I.1 ANALYSIS OF POSITIONAL ENCODING IN INR2JLS

In our proposed method, we employ an RNN-based GNN to process network weights recurrently, effectively imitating the sequential nature of neural network inference. Therefore, we do not use positional encoding to indicate the positional information of layers. To gain an empirical understanding of the importance of positional encoding in our method, we conduct an experiments to assess the impact of positional encoding on our model (Table 10). By adding learnable positional embeddings on the node features of different layers, we find that positional embeddings do not significantly enhance the performance of our INR classification model across the four datasets.

Table 10: Analysis on the effect of positional encoding on INR2JLS framework. The performance is measured by the classification test accuracy (%) of the models on MNIST, FashionMNIST, CIFAR-10 and CIFAR-100 INR datasets.

|  | MNIST | FashionMNIST | CIFAR-10 | CIFAR-100 |
|---|---|---|---|---|
| INR2JLS with positional encoding | 98.6±0.02 | 89.9±0.09 | 73.5±0.04 | 42.7±0.16 |
| INR2JLS (Ours) | 98.6±0.01 | 90.6±0.07 | 73.2±0.28 | 42.4±0.32 |

## I.2 ANALYSIS OF NON-LINEARITY EMBEDDINGS IN INR2JLS

In Kofinas et al. (2024), the authors encode non-linearities as node features. This approach is necessary because they use the update function with shared parameters for nodes in all layers of the neural graph. Consequently, non-linear embeddings are required to specify the activation functions used at different nodes. In contrast, as shown in Equation 4, we use independent GNN layers to process the snapshots at each timestamp. Given the expressivity of GNNs on NNs (refer to Appendix B of Kofinas et al. (2024)), each GNN layer can learn to provide different activation functions. Therefore, we do not need to explicitly embed the activation functions into the graphs. To further support our approach, we conduct an experimental analysis by adding learnable non-linear embeddings to our method, following Kofinas et al. (2024). Table 11 shows that the addition of non-linear embeddings does not significantly affect the performance of the INR classification across all four datasets.

Table 11: Analysis on the effect of non-linearity embeddings on INR2JLS framework. The performance is measured by the classification test accuracy (%) of the models on MNIST, FashionMNIST, CIFAR-10 and CIFAR-100 INR datasets.

|  | MNIST | FashionMNIST | CIFAR-10 | CIFAR-100 |
|---|---|---|---|---|
| INR2JLS add non-linearity embeddings | 98.4±0.16 | 90.4±0.01 | 73.2±0.12 | 42.6±0.04 |
| INR2JLS (Ours) | 98.6±0.01 | 90.6±0.07 | 73.2±0.28 | 42.4±0.32 |

