# OpenReview forum: "Dynamic Neural Graph: Facilitating Temporal Dynamics Learning in Deep Weight Space"
_ICLR.cc/2025/Conference — Submitted to ICLR 2025_

### Official Review · Reviewer_yXfK · 2024-10-30

**Soundness:** 2
**Presentation:** 2
**Contribution:** 2
**Rating:** 5
**Confidence:** 2

**Summary:**

The paper studies the problem of using neural networks as implicit neural representations (INRs). To address the overlooking of the sequential nature between neural layers, the authors propose representing neural network parameters from a dynamic graph perspective. Based on this core idea, the paper develops Dynamic Neural Graph Encoder and INR2JLS. Finally, the authors conduct experiments on various tasks to verify the proposed method.

**Strengths:**

- The paper studies INRs from a dynamic graph perspective, which is a new perspective.

- The paper is well-structured.

**Weaknesses:**

- Although the paper uses a new perspective (dynamic graph) to study INRs, regarding the weight parameters as dynamic graphs may not be an effective approach. The neural graph between different layers changes significantly, making it challenging to capture their sequential and evolving characteristics accurately.

- Are there any experiments that prove the improvement introduced by capturing the sequential nature between layers?

- In the related work section, the paper lacks a deeper analysis to describe differences between the proposed method and static graph counterparts.

- The proposed method can regard transformers as dynamic neural graphs, but there lack experiments on the transformers architecture to validate the proposed method.

- Minor typos: “dynmaic” in lines 181, 663, and 691 should be “dynamic”

**Questions:**

See above

---

> ### Author Response · Authors · 2024-11-20
>
> Thank you for your valuable feedback and constructive comments. We address your concerns as below:
>
> ---
>
> > W1&2. Treating weight parameters as dynamic graphs may not be effective, as the neural graph structure between layers changes significantly, making it difficult to accurately capture their sequential and evolving characteristics. Are there any experiments that prove the improvement introduced by capturing the sequential nature between layers?
>
> We thank for your question. In fact, we dedicated a significant portion of the text to explaining how our method captures the sequential nature of neural network processing, both theoretically and empirically.
>
> - Theoretical analysis: We kindly refer the reviewer to Section 2.3 and Appendix D, where we provide a detailed theoretical analysis of static and dynamic neural graphs, demonstrating the expressiveness of neural networks in these contexts. These analyses demonstrate that our approach can effectively simulate the forward pass of input neural networks, capturing their inherent sequential processing nature. In contrast, the static counterpart may introduce challenging ill-posed problems, leading to a failure in approximating the functionality beyond the first MLP layer.
>
> - Empirical analysis: We kindly refer the reviewer to Appendix E.1, where we include an experimental analysis to further compare the two approaches. These empirical results show that the method  based on static neural graphs is primarily effective for simulating
> only the initial forward pass of the input neural network. In contrast, our proposed dynamic neural graph framework is capable of accurately simulating all forward pass steps of the input neural network.
>
>
> > W3. In the related work section, the paper lacks a deeper analysis to describe differences between the proposed method and static graph counterparts.
>
> At the time of submission, the only work we know employing static graphs was Kofinas et. al. To highlight our primary contribution, we conducted an in-depth study, which is presented in detail in Section 2.3 of the main body. We kindly direct the reviewer’s attention to Section 2.3, where we thoroughly discuss the main limitations of static neural graphs.
>
> > W4. There lack experiments on the transformers architecture to validate the proposed method.
>
> We would like to clarify that our primary contribution does not include applying the proposed method specifically to transformer architectures. While we acknowledge the potential of interpreting transformers as dynamic neural graphs (DNGs), this aspect was only discussed in the Appendix G.2 as a supplementary demonstration to illustrate the flexibility of our framework. It was intended as a conceptual reference rather than a core experimental validation.
>
> To address your concern, we conducted an experiment to predict the generalization of transformers by processing their parameters, thereby validating the effectiveness of our proposed DNG framework in handling transformer architectures.
>
> - Dataset: We follow Small CNN Zoo  (Unterthiner et al.) to prepare a transformer dataset. Specifically, we trained 10,000 differently initialized Vision Transformer (ViT) models (used SimpleViT from ViT_pytorch library) to classify the CIFAR-10 dataset. Each model was trained up to a certain epoch before being stopped, and its parameters and test accuracy were saved. Among the trained models, 80% were used as the training set, 10% as the validation set, and 10% as the test set.
>
> - Implementaion details: Similar to the settings for predicting generalization of CNNs discussed in Section 6.3, we use our proposed DNG-Encoder followed by an MLP to predict the test accuracy of a ViT model given its parameters as input. To construct the DNG, we follow the methods outlined in Appendix G to convert the ViT.
>
> The table below summarizes the running time, memory usage, GFLOPs, and the Kendall rank correlation coefficient τ  of our method and NG-GNN (Kofinas et al.) on the test set.
>
> |                     | Running Time (s) (20 times average) | Memory (MB) | Comp. Cost (GFLOPs) | Kendall rank correlation coefficient τ |
> |---------------------|-------------------------------------|-------------|----------------------|---------------------------------------|
> | NG-GNN           | 0.08524                           | 19.45    | 2.56         | 0.8814 ± 0.002                        |
> | Ours          | 0.01422                            | 21.30    | 1.83         | 0.9004 ± 0.002                        |
>
> > W5. Minor typos.
>
> We have updated our paper to address and correct these typos.

---

> > ### Comment · Reviewer_yXfK · 2024-11-21
> > **Response to Authors**
> >
> > Thank you for your response. Your reply has addressed my concerns. I will raise my score.

---

> > > ### Author Response · Authors · 2024-11-21
> > >
> > > We are glad to hear that our rebuttal has addressed your concerns. Given this, we kindly ask you to consider raising your rating to a positive level, as the current rating remains categorized as negative. We believe this adjustment would align with your comments and affirm the contribution of our paper to the field.  If you have any additional concerns, we would be more than willing to address them during the discussion phase to further improve our submission.

---

> > > > ### Author Response · Authors · 2024-11-25
> > > >
> > > > Dear Reviewer yXfK,
> > > >
> > > > We thank you again for taking the time to thoroughly review our submission and for acknowledging that we have addressed all your concerns. We greatly appreciate your constructive feedback, which has helped us improve the clarity and quality of our work.
> > > >
> > > > Given that we have resolved the issues you raised, we kindly ask if you would consider revising your rating slightly upwards, as it is currently just below the acceptance threshold.
> > > >
> > > > We sincerely respect your perspective and value your judgment, and we would be grateful for any further feedback or suggestions you may have to strengthen the paper even further.
> > > >
> > > > Thank you again for your time, effort, and thoughtful consideration. We deeply appreciate your contributions to the review process.
> > > >
> > > > Best regards,
> > > >
> > > > Authors of Paper 5495

---

> > > > > ### Comment · Reviewer_yXfK · 2024-11-26
> > > > >
> > > > > Thank you for your reply. I maintain my score.

---

### Official Review · Reviewer_dPkK · 2024-10-31

**Soundness:** 3
**Presentation:** 3
**Contribution:** 2
**Rating:** 5
**Confidence:** 3

**Summary:**

This paper proposes a novel approach to model neural networks as dynamic graphs for capturing layer-by-layer dependencies. The authors introduce an RNN-based method named the Dynamic Neural Graph Encoder (DNG-Encoder) for capturing temporal dynamics, which can mirror forward passes and preserve sequential characteristics of neural networks. Additionally, the authors present a new framework named INR2JLS, which can map neural weights and data into a unified latent space for enhancing the quality of representations. Extensive experiments are conducted to demonstrate the effectiveness and significant improvements of the proposed method.

**Strengths:**

- Different from traditional methods, this paper introduces a novel appoach (The Dynamic Neural Graph Encoder, DNG-Encoder) to model neural networks as dynamic graphs, effectively capturing temporal dependencies across layers and providing more accurate representations for forward passes of neural networks.
- The INR2JLS framework proposed in this paper can map neural weights and data into a unified latent space, which can enhance the quality of representations and improve the model performance particularly for challenging applications like implicit neural representation (INR) classification.

**Weaknesses:**

- In Section 3.1, the authors build on the work proposed by [1] and suggest that the natural symmetries in graphs align with neuron permutation symmetries in neural networks. However, as this paper focus on dynamic graphs, differing from the static graph setting in [1], the claim of invariance or equivariance to permutation symmetries requires further proof in the context of dynamic graphs.

- In Section 4.1, the authors use an RNN-based method to model the dynamic behaviors of neural networks; however, gradient vanishing and explosion are common issues in RNN-based methods. Specifically, as the size of neural networks or graphs increases, the dynamic model need to operate over more timesteps, increasing the likelihood of these issues occurring. It would be valuable to explain how the proposed methods how to address these two challenges, supported by a theoretical analysis.

- In Section 7, the authors provide only the experimental results on computational complexity. A more detailed theoretical analysis of time and space complexity should be included for fair comparisons with baseline methods [1], [2], and [3].

[1] Kofinas M, Knyazev B, Zhang Y, et al. Graph Neural Networks for Learning Equivariant Representations of Neural Networks[C]//The Twelfth International Conference on Learning Representations.

[2] Zhou A, Yang K, Jiang Y, et al. Neural functional transformers[J]. Advances in neural information processing systems, 2024, 36.

[3] Zhang D W, Kofinas M, Zhang Y, et al. Neural networks are graphs! graph neural networks for equivariant processing of neural networks[J]. 2023.

**Questions:**

- Please provide further theoretical proof of invariance or equivariance to permutation symmetries within the context of dynamic graphs.

- Please explain how the proposed method addresses gradient vanishing and explosion issues, both experimentally and theoretically.

- Please demonstrate the scalability of the proposed method, showing its deployment on large-scale neural networks and graphs.

- Can the proposed methods be applied to other types of neural networks?

---

> ### Author Response · Authors · 2024-11-20
>
> Thank you for your valuable feedback and constructive comments. We address your concerns as below:
>
> ---
>
> > W1. The claim of invariance or equivariance to permutation symmetries requires further proof in the context of dynamic graphs.
>
> To address your concern, we have revised our paper and added a detailed theoretical analysis, which can now be found in Appendices B and C.
>
> > W2. It would be valuable to explain how the proposed methods address these gradient vanishing and explosion problems.
>
> In our method, we utilize Gated Recurrent Units (GRUs) as the recurrent memory updating function. GRUs are specifically designed to address the issues of gradient vanishing and explosion, which commonly affect standard RNNs, particularly when processing long sequences. These issues can be potentially mitigated by the gating mechanisms in GRUs, which effectively regulate the flow of information and gradients.
>
> As per the reviewer fjzh’s request, we conducted additional experiments by replacing the GRUs with a standard RNN. We observed a performance difference, with GRUs outperforming RNNs. Naïve RNNs are well-known to gradient vanishing/explosion problems. However, we cannot conclusively attribute this performance discrepancy to gradient vanishing or explosion. During the training process of both networks, we monitored the gradient norms and found that, in both cases, the gradients remained within a reasonable range throughout the training.
>
> We acknowledge that gradient vanishing or explosion may occur in larger networks. However, this field is still in its early stages of development, with relatively small datasets. In this paper, we have made further attempt by introducing the classification task on larger dataset, i.e., CIFAR-100-INR. Even in this classification dataset, most existing methods performed poorly. In contrast, our approach demonstrated a significant improvement, improving over the state-of-the-art by 10.75%. We hope that as the field progresses, gradient vanishing and explosion issues will become more evident, and addressing these challenges will be a key focus of our future work.
>
> > Q1. Can the proposed methods be applied to other types of neural networks?
>
> Currently, our method supports processing structures such as MLPs, CNNs, and Transformers. For other types of networks, further research and adjustments would be needed to adapt our approach. We would also greatly appreciate it if the reviewer could suggest specific types of neural networks they are interested in exploring with our method.
>
> > W3. A more detailed theoretical analysis of time and space complexity should be included.
>
>
> Thank you for your question. We chose not to provide a theoretical time and space complexity analysis primarily because such analyses often may not accurately reveal the deployment cost in practice. For example, in Kofinas et al., the authors may use the PNA backbone but significantly increase the parameters of $\phi_m$, the message function, to improve performance. In such cases, the overall computational complexity could be dominated by the larger message function. Similarly, many modern machine learning competitions prioritize real-world computational costs on specific datasets rather than theoretical estimates.
> This is why we believe the empirical results presented in Table 6 more effectively reflect the practical computational costs of different methods.
>
> To further address the reviewer's concern, we conducted a computational complexity analysis of the mentioned papers, which is detailed below. Please note that the theoretical results for Kofinas et al. are derived by us, as we are unable to find any explicit theoretical analysis of time and space complexity in their paper. The computational complexity analysis of our method can now also be found in Appendix F.

---

> ### Author Response · Authors · 2024-11-20
>
> ## Cont.
> ## Theoretical Results: Computational Cost
>
> To address the reviewer's concern, we present our theoretical results on the computational cost of the mentioned methods. We omit an analysis of [3] because reference [3] is the workshop version of reference [1]. Since neither our method nor [1,3] requires a decoder for inference, this analysis focuses on providing theoretical results for the computational costs within the encoder.
>
> To determine the time complexity of applying an L-layer MPNN on a graph with the following characteristics:
> - **Number of nodes per MLP layer:** $n$. For simplicity, we assume each MLP layer has $n$ neurons.
> - **Dimension of node/edge features:** $d$. For simplicity, we assume the dimension of edge and node features are the same.
> - **Number of MLP layers:** $L$.
> - **Number of MPNN layers:** $L$. In [1], the number of MPNN layers is set to be the same as the number of MLP layers.
> ---
> ### * Time Complexity of Our Method
> 1. **Message Computation (Equation 4):**
>    - Per edge computation: $O(d^2)$ (including edge feature transformation).
>    - Total edges computation: $O(n^2 \cdot L \cdot d^2)$. There are $n^2$ edges in an MLP layer and $L$ total layers.
>
> 2. **Aggregation (Equation 4) :**
>    - Each node aggregation: $O(n \cdot d)$. Each node aggregates messages from its $n$ neighbors in the graph of previous time step.
>    - Total nodes: $O(n^2 \cdot L \cdot d)$.
>
> 3. **Recurrent Memory Updates (Equation 6):**
>    $O(n \cdot L \cdot d^2)$.
>
> **Total Computational Complexity:**  $O(n^2 \cdot L \cdot d^2 + n^2 \cdot L \cdot d + n \cdot L \cdot d^2) $
>
> *Analysis:* For large input networks, $O(n^2 \cdot L \cdot d^2)$ dominates.
>
>
>
> ### * Space Complexity of Our Method
> 1. **Node and Edge Features:** $O(n^2 \cdot d + n \cdot d)$. We only store the node and edge features of the graph from the previous time step, corresponding to the previous MLP layer.
> 2. **Memory (Equation 6):** $O(n \cdot d)$.
>
> **Total Space Complexity:**  $ O(n^2 \cdot d + n \cdot d) $
>
> ---
>
> ### *Time Complexity of [1,3]
> 1. **Message Computation:**
>    - Per edge computation: $O(d^2)$ (including edge feature transformation).
>    - Total edges computation: $O(n^2 \cdot L^2 \cdot d^2)$. There are $n^2$ edges in an MLP layer and $L$ total layers. The same computation is executed $L$ times since they have an $L$-layer MPNN.
>
> 2. **Aggregation:**
>    - Each node aggregation: $O(n \cdot d)$.
>    - Total nodes: $O(n^2 \cdot L^2 \cdot d)$.
>
> 3. **Node Updates:** $O(n \cdot L^2 \cdot d^2)$.
>    Each node update costs $O(d^2)$, and there are $n \cdot L^2$ updates.
>
> 4. **Edge Updates:** $O(n^2 \cdot L^2 \cdot d^2)$.
>    Each edge update costs $O(d^2)$, and there are $n^2 \cdot L^2$ updates.
>    *Note:* We do not update edges.
>
> **Total Computational Complexity:**  $O(n^2 \cdot L^2 \cdot d^2 + n^2 \cdot  L^2 \cdot d + n \cdot L^2 \cdot d^2) $
>
> *Analysis:* For large input networks, $O(n^2 \cdot L^2 \cdot d^2)$ dominates.
>
> ### *Space Complexity of [1,3]
> 1. **Node and Edge Features:** $O(L \cdot n^2 \cdot d + n \cdot d)$.
>    NG needs to store the node and edge features of the whole static graph.
>
> **Total Space Complexity:**  $O(L \cdot  n^2 \cdot d + n \cdot d)$
>
> ---
>
> ### *Computational Complexity of [2]
> In the algorithm of [2], both the weights and biases of the original input network are updated. Since the number of weights is significantly larger than biases, we calculate complexity based only on the operations on the network weights:
> 1. **Self-Attention Layer Computation:** $O(L \cdot n^3 \cdot d)$.
> 2. **MLP Computation:** $O(L \cdot n^2 \cdot d^2)$.
>
> **Total Computational Complexity:**
> $ O(L \cdot n^3 \cdot d + L \cdot n^2 \cdot d^2) $
>
> ### *Space Complexity of [2]: $ O(n^2 + n \cdot d) $.
> This is similar to the cost in transformer.
>
> ---
>
> ### Summary:
>
> | **Method**  | **Time Complexity**                             | **Space Complexity**                 |
> |-------------|------------------------------------------------|--------------------------------------|
> | **Ours**    | $O(n^2 \cdot L \cdot d^2 + n^2 \cdot L \cdot d + n \cdot L \cdot d^2)$ | $O(n^2 \cdot d + n \cdot d)$       |
> | **[1,3]**   | $O(n^2 \cdot L^2 \cdot d^2 + n^2 \cdot L^2 \cdot d + n \cdot L^2 \cdot d^2)$ | $O(L \cdot n^2 \cdot d + n \cdot d)$ |
> | **[2]**     | $O(L \cdot n^3 \cdot d + L \cdot n^2 \cdot d^2)$ | $O(n^2 + n \cdot d)$               |
>
> In this analysis, we have demonstrated that our method achieves superior performance compared to the alternatives.  Specifically, when compared with the static neural graph [1,3],  the dominant term in our time complexity, $O(n^2 \times L \times d^2)$, is significantly lower than the corresponding term in [1,3], which scales as $O(n^2 \times L^2 \times d^2)$. Additionally, our space complexity, $O(n^2 \cdot d + n \cdot d)$, is more resource-efficient than the $O(L \times n^2 \cdot d + n \cdot d)$ requirement of [1,3].

---

> > ### Author Response · Authors · 2024-11-25
> >
> > Dear Reviewer dPkK,
> >
> > We provided a detailed rebuttal to your question on November 20th, but we have not yet received your feedback. As there are only two days remaining for the discussion phase, we wanted to kindly follow up to ensure there is sufficient time to address any further concerns or clarifications you may have.
> >
> > We deeply value your insights and suggestions, which have been instrumental in refining our work. Please let us know if there is any additional information or clarification you require from our side.
> >
> > Thank you once again for your time and thoughtful feedback. We look forward to hearing from you soon.
> >
> > Best regards,
> >
> > Authors of Paper 5495

---

> > > ### Comment · Reviewer_dPkK · 2024-12-01
> > >
> > > Thank you for your reply. I maintain my score.

---

> > > > ### Author Response · Authors · 2024-12-02
> > > >
> > > > Hope you had a wonderful Thanksgiving! And thank you for sharing your decision with us. We sincerely ask if there are any specific concerns or questions regarding our response that led to your current rating. We would greatly appreciate the opportunity to address them further and provide any necessary clarifications or updates.

---

### Official Review · Reviewer_fjzh · 2024-11-04

**Soundness:** 2
**Presentation:** 2
**Contribution:** 2
**Rating:** 6
**Confidence:** 3

**Summary:**

The paper proposes to model neural network weights as dynamic neural graphs. Such an approach addresses the limitations of previous "static" neural graphs by more closely mimicking the forward pass of MLPs/CNNs and thereby simplifying the task of learning from the weights. The paper proposes to use dynamic GNNs to learn from dynamic neural graphs and shows better results on the tasks such as INR classification and predicting CNN generalization.

**Strengths:**

1. The observation that static neural graphs are not well aligned with the forward pass is original and interesting.
2. The overall idea of using temporal GNNs is logical and novel in this context.
3. The idea of joint weight and image space (INR2JLS) is interesting and novel.
4. The experiments show improvements over the baselines.

**Weaknesses:**

1. The motivation of processing INR weights in the Intro is not convincing. For example, the authors say "This observation has motivated  us to investigate the potential for directly processing INRs to uncover information about the data they encode." It's not very clear uncovering which data the authors imply and why we need to uncover them. It seems that INR classification appeared as a task in the previous literature mainly because it's a convenient testbed for this kind of methods. But recent papers in this domain often add other more practically relevant use-cases (e.g. learning to optimize in Kofinas et al. or processing the weights of diverse transformers in [a]), which makes the motivation of these methods more convincing.
2. The paper [a] (ICLR 2024) is not discussed, however, it proposed an approach very similar to neural graphs of Kofinas et al.
3. As mentioned in 1 above, [a] showed the application to diverse transformer architectures, which could be leveraged in this submission to enhance experiments.
4. Using timestamps is not well justified because the layers in neural networks, while sequential, do not have the notion of time. For example, there is no need to obtain node/edge embedding at continuous times. And usually temporal/dynamic GNNs are used for continuous time prediction. Perhaps, the idea of using timestamps could be more justified for networks such as neural ODEs.
5. More ablations could be done (potentially 3-5 ablations of different model components). For example, is it possible to provide results of INR2JLS with some baseline weight encoders like NFN/NG/etc? Can the authors ablate the GRU (Eq. 6)?


References:

[a] Graph Metanetworks for Processing Diverse Neural Architectures", ICLR 2024

**Questions:**

Regarding Table 4, do the baselines use any augmentation? If not, is comparison in Table 1 fair?

Does the number of heads in multi-head message function needs to be predefined before training DNG-Encoder? Does it mean once it's trained, it cannot be applied to CNNs with larger kernels? Does any of the experiments in the paper have a task where the GNN has to generalize to larger kernels?
Any difference between multi-head message function and using "towers" from MPNN Gilmer et al. (2017)?

---

> ### Author Response · Authors · 2024-11-20
>
> Thank you for your valuable feedback and constructive comments. We address your concerns as below:
>
> ---
> > W1. The motivation for processing INR weights in the Intro is unconvincing. INR classification seems to have been used previously as a convenient testbed, while recent works add more practical use-cases.
>
> **The Motivation:**
>
> Thank you for raising this concern. We agree that the motivation of "uncovering information about the data they encode" aligns more closely with tasks such as classifying and editing INRs. In light of this, we have revised the motivation in our paper to reflect a broader perspective. Please refer to Lines 29-33 for the updated version.
>
> **The Value of INR Classification:**
>
> We argue that INR classification is not merely a testbed for evaluating methods, but rather has potential practical value in real-world applications.  For example, recent work [1] demonstrates that INRs exhibit great potential as a compressed representation for images. Using INRs to store images can yield improved rate-distortion performance, outperforming commonly used formats like JPEG2000. This trend suggests that future image compression protocols may involve transmitting INRs between transmitters and receivers. In this context, at the receiver side, upon receiving the INR, an additional step is required to reconstruct the original image for human viewing. Interestingly, another line of research [2] points out that much of the captured visual content may not be intended for human perception, but rather for automated machine vision analytics. For example, given an image, machine may only be interested in determining whether it contains a dog or a cat, rather than reconstructing the original pixel-level content. This is essentially a classification problem. Inspired by this, when we receive an INR, engineers can adopt the method developed in our paper to directly classify the INR, eliminating the need to reconstruct the image.
>
> **Comparison with Kofinas et al. and Lim et al. [a] on Additional Two Tasks:**
>
> Thanks for the suggestion. We did not include experiments related to “learning to optimize” because the editing of INRs in our experiment demonstrates the capability to modify the weight space and transform the functionality of neural networks. In this task, our method outperforms Kofinas et al.'s approach by a significant margin, achieving an average improvement of over 50% in terms of MSE loss. Additionally, we apologize for not being able to complete the comparison within the rebuttal period, due to the unavailability of their code and dataset for the task. We plan to update the results in the final version of our paper.
>
> Regarding the experiments in Lim et al. [a], although they investigate multiple architectures, their primary tasks are similar to those in our paper, including predicting neural network accuracy and editing INRs. While we intended to perform the comparison, we unfortunately found that both their code and the "diverse architecture" dataset are not publicly available. Given the time required to construct the datasets and reproduce their results, we will include an update on these experiments in the final version of our paper.
>
> [1] Strümpler, Yannick, et al. "Implicit neural representations for image compression." European Conference on Computer Vision. Cham: Springer Nature Switzerland, 2022.
>
> [2] Choi, Hyomin, and Ivan V. Bajić. "Scalable image coding for humans and machines." IEEE Transactions on Image Processing 31 (2022): 2739-2754.
>
>
>
>
> > W2. Lim et al. [a] is not discussed, similar to neural graphs of Kofinas et al.
>
> Thanks for point out this related work that we missed. We have included a discussion of this paper in both the main body and related work section.

---

> > ### Author Response · Authors · 2024-11-20
> >
> > > W3. Lim et al. [a] showed the application to diverse transformer architectures, which could be leveraged in this submission to enhance experiments.
> >
> > Although we cannot perform experiments on the diverse transformer architectures proposed in Lim et al. [a] due to the unavailability of both code and dataset, we conducted an experiment to predict the generalization of transformers by processing their parameters, thereby validating the effectiveness of our proposed DNG framework in handling transformer architectures.
> >
> > - Dataset: We follow Small CNN Zoo  (Unterthiner et al.) to prepare a transformer dataset. Specifically, we trained 10,000 differently initialized Vision Transformer (ViT) models (used SimpleViT from ViT_pytorch library) to classify the CIFAR-10 dataset. Each model was trained up to a certain epoch before being stopped, and its parameters and test accuracy were saved. Among the trained models, 80% were used as the training set, 10% as the validation set, and 10% as the test set.
> >
> > - Implementaion details: Similar to the settings for predicting generalization of CNNs discussed in Section 6.3, we use our proposed DNG-Encoder followed by an MLP to predict the test accuracy of a ViT model given its parameters as input. To construct the DNG, we follow the methods outlined in Appendix G to convert the ViT.
> >
> > The table below summarizes the running time, memory usage, GFLOPs, and the Kendall rank correlation coefficient τ  of our method and NG-GNN (Kofinas et al.) on the test set.
> >
> > |                     | Running Time (s) (20 times average) | Memory (MB) | Comp. Cost (GFLOPs) | Kendall rank correlation coefficient τ |
> > |---------------------|-------------------------------------|-------------|----------------------|---------------------------------------|
> > | NG-GNN           | 0.08524                            | 19.45    | 2.56         | 0.8814 ± 0.002                        |
> > |Ours          | 0.01422                            | 21.30    | 1.83         | 0.9004 ± 0.002                        |
> >
> > > W4. Using timestamps is not well justified because the layers in neural networks, while sequential, do not have the notion of time.
> >
> > While it is true that layers in neural networks do not inherently have a notion of time, the introduction of timestamps in the context of dynamic graphs aligns naturally with the sequential inference process of neural networks. Specifically, timestamps help segment the network into distinct stages, mimicking the layer-by-layer processing that occurs during inference.
> >
> > To support this, we demonstrated in Appendix D that the use of the DNG-Encoder to update nodes in a dynamic graph effectively simulates the sequential updating pattern of neural networks. This re-formulation of neural networks as dynamic graphs represents the major novelty of our method. We believe that a well-designed problem formulation has the potential to inspire more advanced research in a given field. We hope our dynamic graph formulation will encourage further work by considering the temporal dynamics of neural networks' processing.

---

> > > ### Author Response · Authors · 2024-11-20
> > >
> > > > W5. More ablations could be done.
> > >
> > > **Ablation of Encoder:**
> > >
> > > Thanks for your suggestion. Based on your feedback, we conducted a new ablation study to compare the performance of different encoders. In this study, to isolate the potential influence of the decoder, we directly attached classifiers to each encoder and trained the models end-to-end. The results of this experiment are summarized in the table below. This results have also been shown in Appendix E.2.
> > >
> > > It is worth noting that the NFN model does not include a decoder; instead, its encoder is extremely heavy, comprising over 10M parameters. In contrast, our DNG-Encoder, which has a comparable parameter size to NG-GNN and NG-T, significantly outperformed both of these models. Moreover, despite having far fewer parameters than NFN, our method still achieves better performance.
> > >
> > > | Method              | #Params   | MNIST          | FashionMNIST    | CIFAR-10       |
> > > |-------|-------------|-------------|-------------|-------------|
> > > | NFN             | ~135M     | 92.9 ± 0.38     | 75.6 ± 1.07     | 46.6 ± 0.13   |
> > > | NG-GNN             | ~0.3M     | 79.6 ± 1.3     | 71.1 ± 0.42     | 43.94 ± 0.06   |
> > > | NG-T               | ~0.4M     | 83.43 ± 0.12   | 72.13 ± 0.51    | 44.69 ± 0.03   |
> > > | **DNG-Encoder (ours)** | ~0.4M     | **96.6 ± 0.04** | **78.4 ± 0.61** | **54.0 ± 0.07** |
> > >
> > > **Ablation of GRU: **
> > >
> > > Since the GRU structure is not proposed by us and is not a primary focus of this paper, we utilized it solely to process the sequential DNG. However, we understand that the reviewer might be interested in exploring how the choice of different RNN-like structures could affect the final performance.
> > >
> > > To address this, we conducted additional experiments where we replaced the GRU with a naïve RNN. The results are presented below. It can be observed that using GRU achieves better performance compared to using a naïve RNN. These findings further suggest that our method could potentially achieve even better results by integrating more advanced building blocks.
> > >
> > > | Model | MNIST       | Fashion     | CIFAR10     |
> > > |-------|-------------|-------------|-------------|
> > > | Using GRU   | 98.6±0.01   | 90.6±0.07   | 73.2±0.28   |
> > > | Using RNN   | 98.6±0.05   | 88.8±0.03   | 69.7±0.08   |
> > >
> > > > Q1. Regarding Table 4, do the baselines use any augmentation? If not, is comparison in Table 1 fair?
> > >
> > > We appreciate your question. To ensure a fair comparison,  all our evaluations in Table 1 are conducted on the same dataset with augmentation, based on our implementation.
> > >
> > > > Q2. Does the number of heads in the multi-head message function need to be predefined? If so, does this imply it cannot generalize to CNNs with larger kernels once trained? Do any of the experiments in the paper test the GNN's ability to generalize to larger kernels?
> > >
> > > We confirm that the number of heads in the multi-head message function is a hyperparameter that must be predefined before training the DNG-Encoder. As mentioned in Lines 233–234, we zero-pad all kernels to a maximum size of $ h^l \times w^l $ within the network. This operation enables a unified representation across different kernel sizes, effectively generalizing to smaller kernels.
> > >
> > > However, since this hyperparameter is predefined, our current framework does not support generalization to larger kernels once the model is trained. It is worth noting that this limitation is not unique to our approach; to the best of our knowledge, current methods in the literature, such as those by Kofinas et al., similarly do not support generalization to larger kernels post-training.  We believe that exploring methods to enable such generalization would be an exciting direction for future research.
> > >
> > > > Q3. Any difference between multi-head message function and using "towers" from MPNN Gilmer et al. (2017)?
> > >
> > > In Gilmer et al., the ‘multiple towers’ method is proposed to address the computational expense that arises when the dimensionality of node embeddings becomes too large. This approach splits a $d$-dimensional node embedding into $k$ smaller embeddings, each with a dimension of $d/k$, processes them separately, and then merges them. In contrast, our multi-head message function primarily aims to ensure that a source node can transmit n distinct messages to a target node through n edges, thereby better simulating the forward propagation process of a convolutional layer. Instead of splitting node embeddings into different parts, we map the source node embedding into $n$ distinct embeddings, allowing each embedding to interact with a specific edge. These interactions produce n heads, which are then merged to complete the computation.

---

> > > > ### Comment · Reviewer_fjzh · 2024-11-22
> > > >
> > > > Thank you for the response. Below are the following up questions.
> > > >
> > > > **Ablation of GRU**
> > > >
> > > > The new results with the RNN are interesting, but my original comment was about removing the recurrent component altogether, e.g. by using just a linear layer or even no layer at all (e.g. s_i could be set equal to m_i + v_i or along this line).
> > > > Also, it is important to give credit to existing works that also perform GRU-style graph traversal (e.g. Directed Acyclic Graph Neural Networks and Graph HyperNetworks for Neural Architecture Search). E.g. the latter paper also defines the "backward" propagation by traversing the graph in the backward direction, which improves representation, so can be utilized in the proposed approach as well.
> > > >
> > > > **Generalizing to larger kernel sizes**
> > > >
> > > > Given that previous works like Kofinas et al. and your submission do not allow for generalization to larger kernel sizes, what's the motivation of changing the way kernels are processed?
> > > >
> > > > **Multi-head message function**
> > > >
> > > > It still remains unclear what's the conceptual difference between the approaches as both split the features and merge them after message passing and both improve efficiency. The argument that the proposed one "better simulating the forward propagation process of a convolutional layer" is not very strong. It would be interesting to compare the results of the two approaches and highlight the difference in the submission.
> > > >
> > > > **SimpleViT experiments**
> > > >
> > > > This experiment is interesting. Are the architectures the same among all 10k models? Is the graph of neuron connectivity (neural graph) the same in your approach and NG-GNN? How the timestamps are defined for parallel branches like heads and q,k,v? Are their timestamps the same if they are in the same transformer layer?
> > > > It's also interesting to know about the efficiency comparison more. Is the method faster than NG-GNN primarily because it does not update edge features? Why the memory consumption is still high?
> > > >
> > > > **Updating edge features**
> > > >
> > > > It looks like the proposed method does not update edge features and so cannot be used for weight reconstruction/generation purposes? Is it correct?

---

> > > > > ### Author Response · Authors · 2024-11-23
> > > > >
> > > > > Dear Reviewer fjzh,
> > > > >
> > > > > Thank you for taking the time to review our paper and provide additional feedback! We address your questions as below:
> > > > >
> > > > > > Ablation of GRU.
> > > > >
> > > > > - Thank you for further clarifying your question. To address it, we followed your suggestion and conducted an INR classification experiment using INR2JLS without incorporating any RNN-like structures to update node memories. Specifically, we simply added $m_i$ and $v_i$ to obtain $s_i$. The results showed that approaches using RNN-like structures for memory updates significantly outperformed the approach without RNNs (using addition). This further highlights the importance of using RNNs for memory updates.
> > > > >
> > > > >
> > > > > | Dataset      | MNIST        | Fashion      | CIFAR10      |
> > > > > |--------------|--------------|--------------|--------------|
> > > > > | Using GRU    | 98.6±0.01    | 90.6±0.07    | 73.2±0.28    |
> > > > > | Using RNN    | 98.6±0.05    | 88.8±0.03    | 69.7±0.08    |
> > > > > | Using Addition          | 66.0±0.15    | 65.4±0.10    | 41.2±0.06    |
> > > > >
> > > > > - We sincerely thank you for bringing these two related works to our attention. In response, we have updated our paper and included a discussion of these works in lines L295–299.
> > > > >
> > > > >
> > > > > > Generalizing to larger kernel sizes.
> > > > >
> > > > > The primary motivation for modifying the kernel processing approach is to ensure that the message passing mechanism aligns more closely with the computational logic of a standard convolutional layer. As detailed in Section 3 of Kofinas et al., their message passing step is described by Equation 4:
> > > > >
> > > > > $
> > > > > m_{ij} = \phi_{\text{scale}}(e_{ij}) \odot \phi_{m}([v_i, v_j]) + \phi_{\text{shift}}(e_{ij}),
> > > > > $
> > > > >
> > > > > where the edge representation $e_{ij}$ undergoes transformations ($\phi_{\text{scale}}$ and $\phi_{\text{shift}}$) and interacts with the node representation through element-wise multiplication and addition.
> > > > >
> > > > > In their implementation, the convolutional kernel is flattened into a vector, which serves as the edge feature. This edge feature is then linearly transformed and element-wise multiplied with the node representation. To achieve the element-wise multiplication between edge and node features, it is necessary to ensure that both have the same dimensionality. Considering that edge features usually have higher dimensions than node features, the linear transformation typically downscales the original edge features to match the dimensionality of the node features, which can lead to information loss. For example, in extreme cases, an edge vector of size $hw$ may need to be downscaled to a  dimension of $1$. In the work of Kofinas et al., to avoid this catastrophic downscaling, they opted to downscale the edge features and upscale the node features to an intermediate dimensionality between the two.
> > > > >
> > > > > However, we argue that this approach still diverges from the core computational principles of a convolutional layer. In a standard convolutional layer, each kernel value is multiplied by the corresponding value in the feature map channel, and the results are subsequently processed.
> > > > >
> > > > > To address this, we propose a method that faithfully replicates the standard convolution operation while maintaining consistency with the computational logic of convolutional layers. The key idea is to allow each edge to undergo a separate interaction with the node representation. Specifically, each kernel element (edge) is first multiplied with the corresponding node element, and the contributions from all edges are then aggregated to produce the final output.

---

> > > > > > ### Author Response · Authors · 2024-11-23
> > > > > >
> > > > > > > Multi-head message function.
> > > > > >
> > > > > > We would like to clarify that we do not split node features. Instead, we map a single set of node features to multiple embeddings. Specifically, given a node $s_j$, we employ several linear layers ($W_1， W_2, ... , W_N$) to map the $s_j$ to $N$ embeddings. Each embedding subsequently interacts with a specific edge. Assuming that this linear mapping does not result in any information loss, we expect that each edge can effectively interact with the entire node features. Conceptually, this is similar to a convolutional kernel sliding across the complete input feature map.
> > > > > >
> > > > > > In contrast, Gilmer et al. propose to divide $s_j$ into $n$ parts ($s_j^1, s_j^2, ... , s_j^n$).  The original node $s_j$ can then be reconstructed by concatenating these parts:
> > > > > > $s_j = \text{concat}(s_j^1, s_j^2, \dots, s_j^n)$.
> > > > > > They then employ separate message and update functions for each part, which implies that each edge can only interact with a small portion of the node features.
> > > > > >
> > > > > > To furtehr demonstrate, we here conduct an additional experiment. Specifically, we adopt the approach proposed by Gilmer et al. (2017) to split each node feature in a convolutional layer into the same number of parts as the edges, allowing each part to interact with a corresponding edge using the same computation as our method, and then merge the outputs. Additionally, we slightly adjusted the node embedding dimension in the model, increasing it from 128 to 144 (the model's parameter count increased from 3.0M to 3.1M), so that it can be evenly divided by the number of edges. We compared this approach with our method on two datasets (CIFAR-10-GS and SVHN-GS) using the Kendall rank correlation coefficient τ as the metric. We found that our approach, which maps the node feature into multiple features, outperforms the approach proposed by Gilmer et al. (2017). A discussion between the two method is presented between L289-294.
> > > > > >
> > > > > > | Method                | CIFAR-10-GS   | SVHN-GS       |
> > > > > > |-----------------------|---------------|---------------|
> > > > > > | Gilmer et al. (2017)  | 0.932±0.001   | 0.852±0.003   |
> > > > > > | Ours                  | 0.936±0.000   | 0.867±0.002   |

---

> > > > > > > ### Author Response · Authors · 2024-11-23
> > > > > > >
> > > > > > > > SimpleViT experiments.
> > > > > > >
> > > > > > > Thanks for the questions. We have updated the supplementary material by including the revised code that transforms the ViT into a dynamic neural graph. Below are our responses to each of your questions:
> > > > > > >
> > > > > > > **Architecture of Transformers:**
> > > > > > >
> > > > > > > We confirm that all 10,000 ViTs share the same structure. We follow the CNN Zoo to randomly initialize these ViTs, and each ViT is trained up to a certain epoch before being stopped. Unfortunately, due to time constraints during the rebuttal period, we were unable to generate a larger dataset. However, we will show our results on a larger dataset in the final version to strengthen this analysis.
> > > > > > >
> > > > > > >
> > > > > > > **Neuron Connectivity:**
> > > > > > >
> > > > > > > The primary difference lies in how the graph is constructed from the multi-head self-attention module. As detailed in Appendix G.2, to transform the multi-head self-attention module into a dynamic neural graph, we split the module into three timestamps, during which we introduced four sets of nodes. In contrast, Kofinas et al. mentioned in Appendix C.4 that NG-GNN transforms the multi-head self-attention module into a neural graph with only three sets of nodes.
> > > > > > >
> > > > > > > Assume the input vectors of the multi-head self-attention module have a dimension of $d_{model}$, with $H$ heads, each of dimension $d_h$. In NG-GNN, the three sets of nodes contain $d_{model}$, $H \times d_h$, and $d_{model}$ nodes, corresponding to the input vectors, the heads, and the outputs, respectively. In contrast, our dynamic neural graph introduces four sets of nodes, containing $d_{model}$, $3 \times H \times d_h$, $H \times d_h$, and $d_{model}$ nodes. These sets of nodes correspond to the input vectors,  the $Q/K/V$,  the heads, and  the outputs, respectively.
> > > > > > >
> > > > > > > The key distinction lies in the additional set of nodes introduced by our method, located between the first and second sets of nodes in NG-GNN. This additional set models the intermediate step in the multi-head self-attention module where input vectors are mapped to $Q$, $K$, and $V$, followed by the scaled dot-product attention operation to generate the heads. In contrast, NG-GNN bypasses this intermediate step, directly connecting the first set of nodes (input vectors) to the second set (heads). In other words, NG-GNN does not explicitly model $Q$, $K$, and $V$, as well as the scaled dot-product attention operation.
> > > > > > >
> > > > > > >
> > > > > > > **Timestamps:**
> > > > > > >
> > > > > > > As is mentioned above, we transform a multi-head self-attention module into three timestamps in our dynamic neural graph. As described in Appendix G.2, the first timestamp simulates the operation of mapping inputs to $Q$, $K$, and $V$; the second timestamp simulates the scaled dot-product attention operation; and the third timestamp simulates the operation of mapping heads to outputs.
> > > > > > > We transform a transformer block into five timestamps in the dynamic neural graph, which included three timestamps for the multi-head self-attention module and two timestamps for the two linear layers in the feed-forward network.
> > > > > > > In our experiments, the ViT model consisted of two transformer blocks (10 timestamps), one embedding layer (1 timestamp) and one output layer (1 timestamp). Therefore, our dynamic graph for the ViT includes 12 timestamps in total.
> > > > > > >
> > > > > > >
> > > > > > > **Efficiency:**
> > > > > > >
> > > > > > > - Yes, the speed improvement over NG-GNN is mainly due to the fact that our method does not update edge features. Besides, NG-GNN adopts many advanced building blocks such as PNA (Corso et al., 2020), the backbone of NG-GNN, and FiLM-GNN (Brockschmidt, 2020). While these methods are effective in improving performance, they also brings additional computations. For example, PNA uses multiple aggregation functions (e.g., mean, max, min, and standard deviation) to capture statistical information from a node's neighborhood. It also incorporates degree-based scalers to normalize and adapt the aggregation to the graph topology. In contrast, we only update memories (nodes), and employs a relatively simple message-passing function.
> > > > > > >
> > > > > > > - As mentioned above, these advanced operations in NG-GNN are usually impose more influence on time complexity, not on the space complexity. However, we have to note that NG-GNN always needs allocate all the node and edge features in the memory, while ours only need to store a small graph at a specific timestamp.  Unfortuanatly, we find that the dominante part is GRU, which maintains hidden states for seuqntial processing and stores additional intermediate results for gate computations. Overall, our memory comsumption is slightly larger than NG-GNN.

---

> > > > > > > > ### Author Response · Authors · 2024-11-23
> > > > > > > >
> > > > > > > > > Updating edge features.
> > > > > > > >
> > > > > > > > No, it is not correct. After recurrently propagating the node and edge features to the final memory using the GRU, we can apply a decoder on the memory to reconstruct or generate the original INRs. We indded have built this method in our paper, between Line 482-502, to verfiy the importance of image reconstruction in the INR2JLS. As is demonstrated, we introduced a method called INR-INR, which leverages the DNG-Encoder to process input INRs. This method then reconstructs the INRs using two MLPs to map the final node memory to the weights and biases of the INRs.

---

> > > > > > > > > ### Author Response · Authors · 2024-11-25
> > > > > > > > >
> > > > > > > > > Dear Reviewer fjzh,
> > > > > > > > >
> > > > > > > > > We greatly appreciate the time and effort you have already dedicated to reviewing our work. However, we have not yet received your feedback regarding the recent response. As there are only two days remaining for the discussion phase, we wanted to kindly follow up to ensure there is sufficient time for any further clarifications or concerns you might have.
> > > > > > > > >
> > > > > > > > > Please let us know if there is anything further we can do to assist or elaborate on.
> > > > > > > > >
> > > > > > > > > Thank you once again for your contributions and guidance.
> > > > > > > > >
> > > > > > > > > Best regards,
> > > > > > > > >
> > > > > > > > > Authors of Paper 5495

---

> > > > > > > > > > ### Comment · Reviewer_fjzh · 2024-11-25
> > > > > > > > > >
> > > > > > > > > > The response of the authors addresses most of my concerns well, so I'm increasing my score to 6.

---

> > > > > > > > > > > ### Author Response · Authors · 2024-11-25
> > > > > > > > > > >
> > > > > > > > > > > Dear Reviewer fjzh,
> > > > > > > > > > >
> > > > > > > > > > > Thank you for increasing your score to a positive level! We greatly appreciate the opportunity to engage in a constructive discussion with you during the rebuttal process. Your valuable feedback and thoughtful insights have been instrumental in helping us strengthen our paper.
> > > > > > > > > > >
> > > > > > > > > > > Thank you once again for your support and consideration.
> > > > > > > > > > >
> > > > > > > > > > > Best regards,
> > > > > > > > > > >
> > > > > > > > > > > Authors of Paper 5495

---

### Meta-Review · Area_Chair_GZJf · 2024-12-19

**Metareview:**

Overall the authors and reviewers have engaged in a good discussion. The authors have been quite active and managed to bump up one reviewer to an acceptance score while two others remained on the fence. All reviewers acknowledge that the paper has valuable contributions but there is also concerns that some claims are not well-justified specifically that dynamical GNNs are well-suited to represent NN weight spaces. This AC has also been over the paper and find it expecting too much prior knowledge from its readers. A saying example is that INR is only spelled out in Section 5 despite being used 161 times throughout the paper. So more work could be done to lay out the argument for the key building blocks of the proposed framework and it be could be made more accessible.

**Additional Comments On Reviewer Discussion:**

None.

---

### Decision · Program_Chairs · 2025-01-22

Reject